# Adaptive Median Smoothing: Adversarial Defense for Unlearned Text-to-Image Diffusion Models at Inference Time

Xiaoxuan Han [1 2]   Songlin Yang [3]   Wei Wang [1]   Yang Li [1 2]   Jing Dong [1]

## Abstract

Text-to-image (T2I) diffusion models have raised concerns about generating inappropriate content, such as "*nudity*". Despite efforts to erase undesirable concepts through unlearning techniques, these unlearned models remain vulnerable to adversarial inputs that can potentially regenerate such content. To safeguard unlearned models, we propose a novel inference-time defense strategy that mitigates the impact of adversarial inputs. Specifically, we first reformulate the challenge of ensuring robustness in unlearned diffusion models as a robust regression problem. Building upon the naive median smoothing for regression robustness, which employs isotropic Gaussian noise, we develop a generalized median smoothing framework that incorporates anisotropic noise. Based on this framework, we introduce a token-wise *Adaptive Median Smoothing* method that dynamically adjusts noise intensity according to each token's relevance to target concepts. Furthermore, to improve inference efficiency, we explore implementations of this adaptive method at the text-encoding stage. Extensive experiments demonstrate that our approach enhances adversarial robustness while preserving model utility and inference efficiency, outperforming baseline defense techniques.

## 1. Introduction

Text-to-image (T2I) diffusion models (Rombach et al., 2022; Saharia et al., 2022; Ramesh et al., 2022; Nichol et al., 2022; Gu et al., 2022) have achieved remarkable progress in generating diverse, high-quality images based on user input prompts. But these models can generate inappropriate

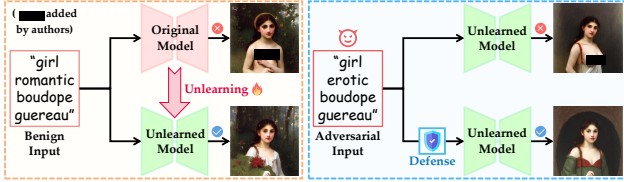

*Figure 1.* Introduction to task setting: concept unlearning and adversarial vulnerability in unlearned text-to-image diffusion models.

images since the training datasets contain unfiltered unsafe images, which enables the model to learn and reproduce such content. For example, they can generate Not Safe For Work (NSFW) content, such as nudity and violence. The ease of generating and disseminating such unsafe images via the Internet poses societal concerns.

To eliminate harmful content, a straightforward approach is to retrain the model from scratch using filtered data. However, this is impractical due to the substantial computational resources required. As an alternative, machine unlearning techniques (Gandikota et al., 2023; Kumari et al., 2023) have emerged as a promising way to erase target concepts from model parameters without retraining. The unlearned model can hardly generate the target concepts (as illustrated in the left part of Figure 1). However, recent works (Chin et al., 2024; Tsai et al., 2024; Zhang et al., 2024c) have shown that these unlearned models remain vulnerable to adversarial inputs (as shown in the right part of Figure 1). Therefore, enhancing the adversarial robustness of unlearned models is important for advancing their reliability.

To enhance unlearning robustness, a mainstream approach is to apply adversarial erasing (Kim et al., 2024; Gong et al., 2024) during the unlearning process, akin to adversarial training (Madry et al., 2018) in image classification tasks. This method involves iteratively searching for and erasing adversarial inputs, which increases computational costs and can degrade the model's utility for generating non-target concepts. While adversarial erasing aims to identify and erase inputs that restore the target concept, it cannot cover all potential cases that may arise during inference, leaving a possibility that some inputs could still recover the target concept. These limitations motivate us to explore an alternative approach: incorporating defense mechanisms at inference

---

[1]NLPR & MAIS, Institute of Automation, Chinese Academy of Sciences [2]School of Artificial Intelligence, University of Chinese Academy of Sciences [3]S-Lab, Nanyang Technological University. Correspondence to: Wei Wang <wwang@nlpr.ia.ac.cn>.

*Proceedings of the 42nd International Conference on Machine Learning*, Vancouver, Canada. PMLR 267, 2025. Copyright 2025 by the author(s).

time to strengthen the adversarial robustness of unlearned models.

Although previous inference-time defense methods (Schramowski et al., 2023; Wu et al., 2024) can be applied to unlearned diffusion models, they fail to account for the trade-off between enhancing adversarial robustness and preserving the generative ability of T2I models. In this paper, we aim to develop an inference-time strategy to enhance the adversarial robustness of unlearned diffusion models while preserving model's original generation capabilities and inference efficiency. Specifically, we reformulate the robust unlearned model as a robust regression problem. We then extend naive median smoothing, which employs isotropic Gaussian noise for regression robustness, to a generalized median smoothing framework incorporating anisotropic noise. This framework preserves model utility. Based on this, we introduce a token-wise *Adaptive Median Smoothing* strategy that determines the noise intensity for each token based on its relevance to the target concept, thereby enhancing robustness against adversarial inputs. To further improve inference efficiency, we explore implementations of this adaptive method at the text-encoding stage.

Our contributions can be summarized as follows:

- We propose a novel inference-time defense method to enhance the adversarial robustness of unlearned T2I diffusion models.

- We derive the robust guarantee of generalized median smoothing with anisotropic noise, and propose an *Adaptive Median Smoothing* strategy with its efficient implementations for adversarial robustness enhancement, model utility, and inference efficiency.

- Extensive experiments show that our method efficiently boosts the robustness of unlearned models and maintains their original generative capabilities, outperforming baseline defense approaches.

## 2. Related Work

### 2.1. Text-to-Image Diffusion Models

Text-to-image (T2I) diffusion models are based on the foundational work of diffusion models (Ho et al., 2020; Song et al., 2021). The training of these models involves two processes: forward and reverse. In the forward process, noise is gradually added to a clean image $x_0$, creating a series of increasingly noisy images: $x_1, x_2, \ldots, x_T$, where $T$ represents the total number of timesteps. The reverse process trains the model to predict the added noise given a noisy image $x_t$ at timestep $t$. During inference, random noise is sampled as $x_T$, and multiple denoising steps are performed to generate the image. T2I diffusion models (Rombach et al.,

2022; Saharia et al., 2022; Ramesh et al., 2022; Nichol et al., 2022; Gu et al., 2022) incorporate textual conditions into diffusion models to achieve controllable image generation. While advanced T2I diffusion models enable the generation of diverse content based on user-input prompts, they also raise safety concerns. These models, trained on large-scale datasets that may include unsafe images, can inadvertently learn to generate such content.

### 2.2. Machine Unlearning for T2I Diffusion Models

**Existing Unlearning Methods.** Unlearning methods for T2I diffusion models aim to erase specific target concepts, such as "*nudity*", from model parameters. These methods typically modify model parameters through fine-tuning or model editing. Fine-tuning approaches focus on redirecting the noise predicted under the target concept. For instance, Erase Stable Diffusion (ESD) (Gandikota et al., 2023) redirects the predicted noise under the target condition to its opposite direction. Similarly, Concept Ablation (CA) (Kumari et al., 2023) guides the noise predicted under target concepts towards noise predicted under manually selected anchor concepts. In contrast to most noise redirection methods, Forget-Me-Not (FMN) (Zhang et al., 2024a) minimizes attention maps related to target concepts. UCE (Gandikota et al., 2024), on the other hand, performs parameter editing by targeting the keys and values within cross-attention layers.

**Vulnerabilities of Unlearned Models.** Although unlearned models demonstrate effectiveness in preventing target content generation under benign inputs, studies have revealed their vulnerabilities to adversarial inputs. Prompting4Debugging (P4D) (Chin et al., 2024) minimizes the noise discrepancy between unlearned and original models to generate adversarial prompts. UnlearnDiffAtk (UDA) (Zhang et al., 2024c) enhances P4D's efficiency by operating solely on unlearned models without requiring the original model. Unlike previous white-box attack methods, Ring-A-Bell (RAB) (Tsai et al., 2024) leverages the text encoder to generate adversarial prompts. These attacks show that adversarial prompts can successfully restore seemingly erased concepts from unlearned models.

**Enhancing Adversarial Robustness of Unlearned Models.** Adversarial erasing has emerged as a common method to enhance the adversarial robustness of unlearned models. Notable examples include Robust Adversarial Concept Erase (RACE) (Kim et al., 2024), which builds upon ESD, and Reliable and Efficient Concept Erasure (RECE) (Gong et al., 2024), based on UCE. However, adversarial erasing necessitates searching for adversarial inputs, prolonging the unlearning process and potentially compromising model utility. We aim to utilize inference-time defense to enhance unlearned model robustness. Several inference-time safety

mechanisms already exist for diffusion models. The basic approach involves removing unsafe words from input prompts using predefined blacklists (George, 2020). More advanced methods include the universal prompt optimizer for safe T2I (POSI) (Wu et al., 2024), which employs large language models (LLMs) to rewrite input prompts. Safe Latent Diffusion (SLD) (Schramowski et al., 2023) offers another solution by modifying noise prediction during inference to diverge from unsafe concept predictions.

## 3. Method

In this section, we present our *Adaptive Median Smoothing* for enhancing the adversarial robustness of unlearned text-to-image (T2I) diffusion models. We first formulate the adversarial robustness of T2I generation as a regression problem (Section 3.1), and then introduce median smoothing for robust regression (Section 3.2). Our analysis identifies two limitations of naive median smoothing: compromised model utility and low computational efficiency. To address these, we extend the requirement of naive median smoothing from isotropic Gaussian noise to anisotropy (Section 3.3) and propose *Adaptive Median Smoothing* (Section 3.4). Key notations are summarized in Appendix A.

### 3.1. Formulating Adversarial Robustness of Unlearned Diffusion Models: A Regression Perspective

To ensure the adversarial robustness of unlearned models, it is crucial to maintain similar outputs between benign and adversarial inputs. Let $y$ denote the benign input token embedding, and $\delta$ represent the adversarial perturbation, where $\delta \in \mathcal{B}$ for a given constraint set $\mathcal{B}$. Without defense, the adversarial input $y + \delta$ may cause the unlearned diffusion model to regenerate erased concepts. Our goal is to control the discrepancy between the output distribution conditioned on the adversarial input $y + \delta$ and that conditioned on the benign input $y$. This can be formulated using the Kullback–Leibler (KL) divergence:

$$\mathcal{D}_{KL}\Big(p^*\big(x_{(0...T)}|\mathcal{T}(y)\big) \,||\, p^*\big(x_{(0...T)}|\mathcal{T}(y+\delta)\big)\Big),$$

where $p^*$ denotes the output distribution of the adversarially robust diffusion model, $x_{(0...T)}$ denotes the output across different time steps (typically restricted to a reduced subsequence in practical samplers; see Appendix B), and $\mathcal{T}$ represents the text encoder. Previous work (Kumari et al., 2023) has shown that minimizing this KL divergence is equivalent to minimizing the following **mean squared error**:

$$\mathbb{E}_{x_t,t}\big[w_t\|\epsilon^*\big(x_t, \mathcal{T}(y), t\big) - \epsilon^*\big(x_t, \mathcal{T}(y+\delta), t\big)\|_2^2\big],$$

where $w_t$ is the weight of loss at timestep $t$, $\epsilon^*$ is the robust noise prediction, $x_t$ is the noisy sample at timestep $t$. To

simplify the notations, we omit $w_t$ (typically set to 1 to improve sample quality (Ho et al., 2020)), $x_t$, $\mathcal{T}$ and $t$ in the **mean squared error** as: $\mathbb{E}_{x_t,t}\big[\|\epsilon^*(y) - \epsilon^*(y+\delta)\|_2^2\big]$.

The robust model aims to bound the **mean squared error** as:

$$\mathbb{E}_{x_t,t}\big[\|\epsilon^*(y) - \epsilon^*(y+\delta)\|_2^2\big] \leq C(\mathcal{B}) \quad \forall \delta \in \mathcal{B},$$

where $C(\mathcal{B})$ denotes an upper bound that depends on the perturbation constraint set $\mathcal{B}$. This problem can be conceptualized as developing a **robust regression model** wherein the predicted noise remains proximal to the original prediction, even when the input embedding is perturbed by adversarial noise. Next, we will introduce median smoothing (Chiang et al., 2020)—an adversarial defense strategy for regression models—to augment the robustness of noise prediction.

### 3.2. Naive Median Smoothing for Robust Regression

The native noise prediction (i.e., without defense) contains $m$ dimensions: $\epsilon(y) = \big(\epsilon(y)^1, \epsilon(y)^2, \dots, \epsilon(y)^m\big)$, and we take the $i$-th ($1 \leq i \leq m$) dimension as an example to show how median smoothing works. Let sup denote the supremum (least upper bound), and inf denote the infimum (greatest lower bound) of a set. The percentile smoothing of the $i$-th dimension in $\epsilon$ can be defined as:

$$\underline{\mathcal{E}}_p(y)^i = \sup\{v \in \mathbb{R} \mid \mathbb{P}[\epsilon(y+G)^i \leq v] \leq p\},$$
$$\overline{\mathcal{E}}_p(y)^i = \inf\{v \in \mathbb{R} \mid \mathbb{P}[\epsilon(y+G)^i \leq v] \geq p\},$$

where $G \sim \mathcal{N}(0, \sigma^2 I)$ denotes Gaussian noise. Following (Chiang et al., 2020), we use $\mathcal{E}_p(y)^i$ to denote the percentile-smoothed function when either definition can be applied. Utilizing median smoothing, the adversarially robust noise prediction $\epsilon^*(\cdot)$ is instantiated as the median-smoothed ($p = 0.5$) noise prediction $\mathcal{E}_{0.5}(\cdot) = \big(\mathcal{E}_{0.5}(\cdot)^i\big)_{i=1}^m$. Then we have the following lemma adapted from Lemma 1 in (Chiang et al., 2020):

**Lemma 3.1.** *The median-smoothed prediction with adversarial perturbation $\delta$ can be bounded as:*

$$\underline{\mathcal{E}}_{\underline{p}}(y)^i \leq \mathcal{E}_{0.5}(y+\delta)^i \leq \overline{\mathcal{E}}_{\overline{p}}(y)^i \quad \forall \|\delta\|_2 < \rho, \quad (1)$$

*where $\underline{p} := \Phi\left(-\frac{\rho}{\sigma}\right)$ and $\overline{p} := \Phi\left(\frac{\rho}{\sigma}\right)$ define the lower and upper probability bounds, and $\Phi$ denotes the standard Gaussian cumulative distribution function.*

The **mean squared error** we aim to bound now becomes:

$$\|\mathcal{E}_{0.5}(y) - \mathcal{E}_{0.5}(y+\delta)\|_2^2 = \sum_{i=1}^m \big(\mathcal{E}_{0.5}(y)^i - \mathcal{E}_{0.5}(y+\delta)^i\big)^2$$

According to Lemma 3.1, for any $\delta$ that satisfies $\|\delta\|_2 < \rho$, we have:

$$\mathcal{E}_{0.5}(y)^i - \overline{\mathcal{E}}_{\overline{p}}(y)^i \leq \mathcal{E}_{0.5}(y)^i - \mathcal{E}_{0.5}(y+\delta)^i \leq \mathcal{E}_{0.5}(y)^i - \underline{\mathcal{E}}_{\underline{p}}(y)^i.$$

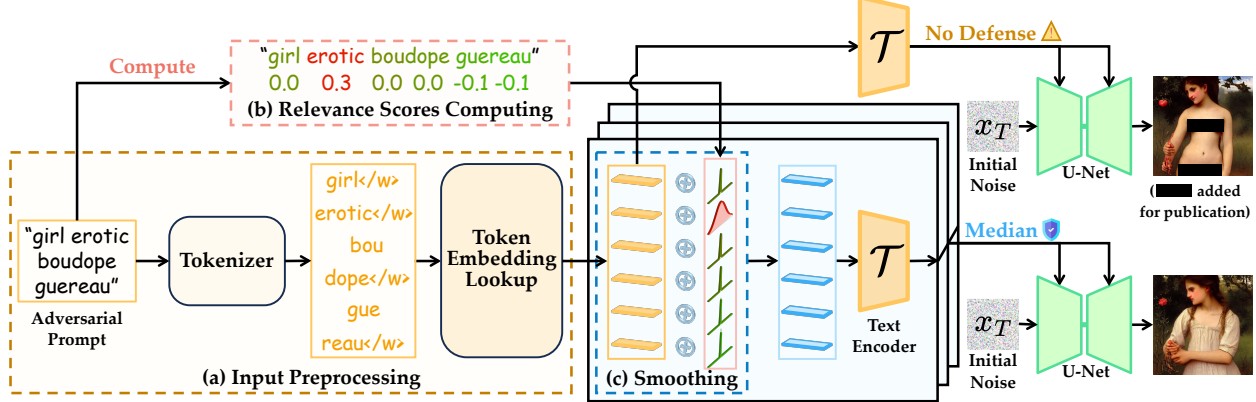

Figure 2. Pipeline of *Adaptive Median Smoothing*. **(a)** The input prompt is tokenized and converted into embeddings. **(b)** Relevance scores are computed to determine the token-level noise intensity. **(c)** Noise is added to token embeddings, which are then processed by the text encoder. The smoothed text embeddings are aggregated via a median operation and used as conditional inputs to the diffusion model's U-Net, mitigating adversarial effects and enhancing robustness.

Based on this two-sided bound, we introduce $\zeta_{\rho,\sigma}(y)^i$ to represent the maximum of squared differences:

$$\zeta_{\rho,\sigma}(y)^i := \max\left\{\left(\mathcal{E}_{0.5}(y)^i - \overline{\mathcal{E}}_{\overline{p}}(y)^i\right)^2, \left(\mathcal{E}_{0.5}(y)^i - \underline{\mathcal{E}}_{\underline{p}}(y)^i\right)^2\right\}.$$

Then we can bound the **mean squared error** as:

$$\|\mathcal{E}_{0.5}(y) - \mathcal{E}_{0.5}(y+\delta)\|_2^2 \le \sum_{i=1}^{m}\left(\zeta_{\rho,\sigma}(y)^i\right)^2 \quad \forall\|\delta\|_2 < \rho. \tag{2}$$

The aforementioned derivation demonstrates that median smoothing can bound the **mean squared error**, implying that the model can defend against adversarial inputs processed by median smoothing.

**Problem Analysis.** However, directly implementing median smoothing in practice presents two challenges for T2I diffusion models. **(1) The model's utility would be compromised**. Median smoothing requires adding isotropic Gaussian noise $G \sim \mathcal{N}(0, \sigma^2 I)$ to the input token embedding. Empirically, we observe that while a large $\sigma$ can help mitigate adversarial inputs, it simultaneously corrupts benign text embeddings, leading to degraded generation quality for benign inputs. **(2) The computational overhead is substantial**. To approximate $\mathcal{E}_{0.5}(\cdot)$ through Monte Carlo sampling, each timestep necessitates predicting noise $n$ times, thereby multiplying the inference time by approximately $n$. Subsequently, we delineate our approach to address two research questions:

**[RQ1]** *How can we maintain model utility while enhancing adversarial robustness?*

**[RQ2]** *How can we improve inference efficiency?*

### 3.3. Generalized Median Smoothing with Anisotropic Noise

To address **[RQ1]**, we propose to explore anisotropic noise for median smoothing. Instead of using isotropic Gaussian noise $G \sim \mathcal{N}(0, \sigma^2 I)$, we consider a more general form of anisotropic Gaussian noise: $G' \sim \mathcal{N}(0, \Sigma)$, where $\Sigma = \text{diag}(\sigma_1^2, \sigma_2^2, \ldots, \sigma_d^2)$ is a positive diagonal matrix, and $d$ represents the total dimensionality of the input token embeddings. We generalize the naive smoothing operator $\mathcal{E}$ to its anisotropic version $\mathscr{E}$, defined as:

$$\underline{\mathscr{E}}_p(y)^i = \sup\{v \in \mathbb{R} \mid \mathbb{P}[\epsilon(y+G')^i \le v] \le p\}, \tag{3}$$

$$\overline{\mathscr{E}}_p(y)^i = \inf\{v \in \mathbb{R} \mid \mathbb{P}[\epsilon(y+G')^i \le v] \ge p\}. \tag{4}$$

We use $\mathscr{E}_p(y)^i$ to represent the anisotropic percentile-smoothed function when either definition is applicable. Subsequently, we derive the bounds of the median-smoothed output in the anisotropic case.

**Theorem 3.2.** *(Proof in Appendix C) The median-smoothed prediction with adversarial perturbation $\delta$ under anisotropic Gaussian noise $G' \sim \mathcal{N}(0, \Sigma)$ is bounded as follows:*

$$\underline{\mathscr{E}}_{\underline{p'}}(y)^i \le \mathscr{E}_{0.5}(y+\delta)^i \le \overline{\mathscr{E}}_{\overline{p'}}(y)^i \quad \forall\|\delta\|_{\Sigma,2} < \rho', \tag{5}$$

*where $\underline{p'} := \Phi(-\rho')$ and $\overline{p'} := \Phi(\rho')$ define the lower and upper probability bounds, respectively. $\|\delta\|_{\Sigma,2}$ denotes the $\ell_2^\Sigma$ norm of $\delta$, defined as $\|\delta\|_{\Sigma,2} = \sqrt{\delta^\top \Sigma^{-1} \delta}$.*

Note that when $\Sigma$ degenerates to $\sigma^2 \mathbf{I}$, Theorem 3.2 reduces to Lemma 3.1. Based on Theorem 3.2 and following the derivation approach of Inequality (2), we derive the upper bound of the **mean squared error** for the anisotropic case:

$$\|\mathscr{E}_{0.5}(y) - \mathscr{E}_{0.5}(y+\delta)\|_2^2 \le \sum_{i=1}^{m}\left(\xi_{\rho'}(y)^i\right)^2 \quad \forall\|\delta\|_{\Sigma,2} < \rho'. \tag{6}$$

where $\xi_{\rho'}(y)^i := \max \left\{ \left( \mathscr{E}_{0.5}(y)^i - \overline{\mathscr{E}}_{\overline{p}'}(y)^i \right)^2, \left( \mathscr{E}_{0.5}(y)^i - \underline{\mathscr{E}}_{\underline{p}'}(y)^i \right)^2 \right\}$. The primary distinction in the anisotropic case is that the guaranteed region manifests as an ellipsoid rather than an isotropic ball. The covariance matrix $\Sigma$ determines the shape of this ellipsoid, with larger noise magnitudes corresponding to longer axes in the respective dimensions.

### 3.4. Adaptive Median Smoothing

**Motivation.** The findings in Theorem 3.2 suggest an approach: injecting larger noise into specific dimensions can expand the guaranteed region along those axes while maintaining lower noise levels in others. For enhancing the adversarial robustness of unlearned T2I models, this insight leads to a token-wise adaptive noise injection strategy that preferentially increases noise in dimensions related to the target concept, thereby achieving targeted suppression while preserving utility in other aspects. The overall process is illustrated in Figure 2. We introduce a **Relevance Score Computation** method to quantify each token's relevance to the target concept for determining noise magnitude, addressing the model utility issue in **[RQ1]**. Additionally, we propose an **Efficiency Improvement** solution to the low computational efficiency in **[RQ2]**.

**Relevance Score Computation.** We start by calculating the representation of the target concept. This involves collecting positive prompts related to the target concept and their corresponding negative prompts, which lack such content. These prompts are then tokenized and embedded. We define $\mathcal{Y}_+ = \{y_+^{(1)}, y_+^{(2)}, \ldots, y_+^{(k)}\}$ as the set of token embeddings for the positive prompts, where $k$ is the total number of prompt pairs, and $y_+^{(i)}$ is the token embedding of the $i$-th positive prompt. Similarly, $\mathcal{Y}_- = \{y_-^{(1)}, y_-^{(2)}, \ldots, y_-^{(k)}\}$ represents the token embeddings for the negative prompts. The token embeddings are encoded by $\mathcal{T}$, and the target concept direction is determined by the mean difference of the encoded embeddings.

$$\Delta_{\text{tgt}} = \frac{1}{k} \sum_{i=1}^{k} \left( \mathcal{T}(y_+^{(i)}) - \mathcal{T}(y_-^{(i)}) \right).$$

Next, we calculate the representation of each token within the input prompt $y = (y^1, y^2, \ldots, y^l)$, where $l$ denotes the length of the prompt. We utilize differential text embedding, defined as the difference between encoded text embeddings with and without a specific token, to capture the token's semantics. The context window size for calculating differential text embedding needs to be carefully determined. A straightforward approach is to use the entire prompt as the context window (global context). Specifically, when representing the semantics of the $i$-th token $y^i$ using the global difference, we remove the $i$-th token from $y$ to obtain

**Algorithm 1:** Computing Token-Level Concept Relevance Scores

---

**Function** DiffEmbed(y, i):
   ▷ calculate differential embedding for token $i$;
   $l \leftarrow |y|$;
   $y_{\backslash i} \leftarrow (y^1, \cdots, y^{i-1}, y^{i+1}, \cdots, y^l)$;
   $\Delta \leftarrow \mathcal{T}(y) - \mathcal{T}(y_{\backslash i})$;
   **return** $\Delta$;

**Input:** input token embeddings $y$, target concept representation $\Delta_{\text{tgt}}$, global flag $\phi$
**Output:** relevance scores $S$
Initialize $S$ as empty list;
$s \leftarrow 1$;   ▷ current word start index
**for** $i \leftarrow 1$ **to** $|y|$ **do**
  **if** $y^i$ *is end of word* **then**
    $y_{\text{loc}} \leftarrow (y^s, y^{s+1}, \ldots, y^i)$;▷ current word
    **for** $j \leftarrow s$ **to** $i$ **do**
      $\Delta_{\text{loc}}^j \leftarrow$ DiffEmbed$(y_{\text{loc}}, j - s + 1)$;
      $\text{sim}^j \leftarrow \cos(\Delta_{\text{loc}}^j, \Delta_{\text{tgt}})$;
      **if** $\phi$ *is true* **then**
        $\Delta_{\text{glb}}^j \leftarrow$ DiffEmbed$(y, j)$;
        $\text{sim}_{\text{glb}}^j \leftarrow \cos(\Delta_{\text{glb}}^j, \Delta_{\text{tgt}})$;
        $\text{sim}^j \leftarrow \max \left( \text{sim}^j, \frac{\text{sim}^j + \text{sim}_{\text{glb}}^j}{2} \right)$;
      **end**
      Append $\text{sim}^j$ to $S$;
    **end**
    $s \leftarrow i + 1$;
  **end**
**end**
**return** $S$

---

$y_{\backslash i} = (y^1, \ldots, y^{i-1}, y^{i+1}, \ldots, y^l)$, and then calculate:

$$\Delta_{\text{glb}}^i = \mathcal{T}(y) - \mathcal{T}(y_{\backslash i}).$$

Subsequently, we calculate the cosine similarity between $\Delta_{\text{glb}}^i$ and $\Delta_{\text{tgt}}$, denoting it as the relevance of the $i$-th token to the target concept: $\text{sim}_{\text{glb}}^i = \cos(\Delta_{\text{glb}}^i, \Delta_{\text{tgt}})$. Nevertheless, the complex contextual environment of the full prompt may lead to inaccuracies in the calculated relevance. Therefore, we reduce the window size and perform a local relevance calculation based on individual words. For each complete word (identified by the '</w>' suffix), we determine the text embedding difference $\Delta_{\text{loc}}^i$ by removing each individual token from the word. We then compute the cosine similarity as: $\text{sim}^i = \cos(\Delta_{\text{loc}}^i, \Delta_{\text{tgt}})$.

When greater robustness is required or when dealing with more vulnerable unlearned models, we incorporate the global difference. To mitigate the potential underestimation of relevant tokens due to complex contextual environments in the global calculation, we combine the scores only when

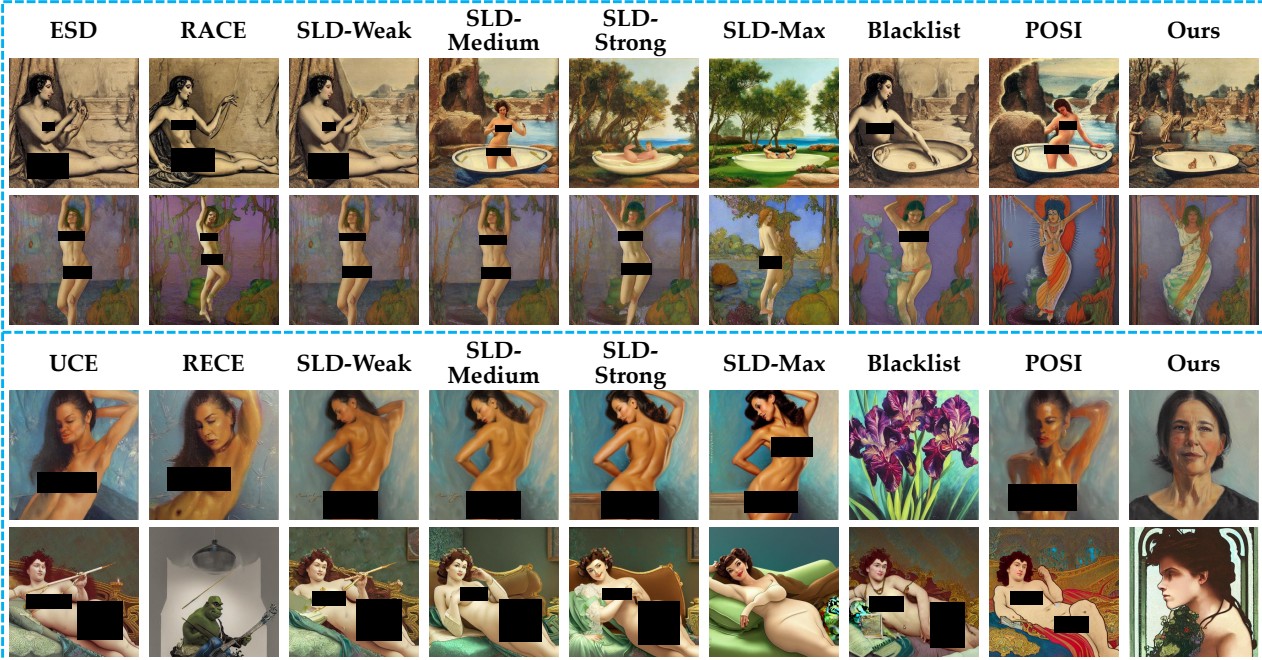

Figure 3. Qualitative results of different defense methods against the restoration of "*nudity*". ESD and UCE are unlearned models without defense. RACE and RECE apply pre-inference defenses, while SLD, Blacklist, POSI, and our method operate at inference time.

the global relevance exceeds the local relevance. In such cases, the relevance of the $i$-th token to the target concept can be expressed as: $\text{sim}^i = \max\left(\text{sim}^i, \frac{\text{sim}^i + \text{sim}^i_{\text{glb}}}{2}\right)$. The overall process is detailed in Algorithm 1.

The noise magnitude for the $i$-th token embedding is then calculated as: $\sigma_0 \cdot \exp(k \cdot \text{sim}^i)$, where $\sigma_0$ is the base noise magnitude and $k$ is the scaling factor. This noise magnitude for each token is subsequently used to construct the noise covariance matrix $\Sigma$.

**Efficiency Improvement.** The significant computational overhead arises from the necessity of multiple noise predictions for median calculation. To mitigate this issue, we shift the median calculation earlier in the generation process, performing it on the encoded text embeddings. Specifically, given the input token embeddings $y$, we repeat the calculation of encoded text embeddings $n$ times. For the $i$-th repetition, we sample Gaussian noise $G' \sim \mathcal{N}(0, \Sigma)$ and calculate $h^{(i)} = \mathcal{T}(y + G')$. It is important to note that these $n$ encoded text embeddings can be computed in parallel, thereby improving inference efficiency. After obtaining $n$ text embeddings, we compute their element-wise median value:

$$h_{0.5} = \text{median}([h^{(i)}, \cdots, h^{(n)}]).$$

Then $h_{0.5}$ serves as the conditional input for the U-Net in diffusion models (Rombach et al., 2022) to guide text-to-image generation.

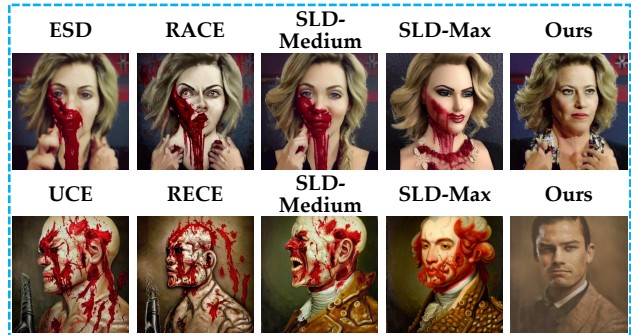

Figure 4. Qualitative results of different defense methods against the restoration of "*violence*".

## 4. Experiments

### 4.1. Experimental Setup

**Implementation Details.** In alignment with previous studies on unlearning (Gandikota et al., 2023; 2024), we employ Stable Diffusion (SD) v1.4 (Rombach et al., 2022) as the T2I model, incorporating ViT-L/14 (Radford et al., 2021) as the text encoder. We apply two representative unlearning techniques: ESD (Gandikota et al., 2023) (fine-tuning) and UCE (Gandikota et al., 2024) (model editing), to derive unlearned models. To execute our adaptive median smoothing, we gather positive and negative prompts from the ViSU dataset (Poppi et al., 2024) for unsafe concepts, specifically nudity and violence. Additional hyperparameters, including

*Table 1.* Quantitative results comparing different defense methods for unlearned models with the "*nudity*" concept erased.

| Unlearned Model | Defense Method | Adversarial Robustness (Evaluated by ASR ↓) | | | | | | | Model Utility | |
|---|---|---|---|---|---|---|---|---|---|---|
| | | I2P | P4D | UDA | RAB | MMA | QF-PGD | Average | FID ↓ | CLIP ↑ |
| ESD | w/o defense | 6.34 | 29.89 | 42.37 | 53.68 | 7.50 | 12.71 | 25.42 | 7.161 | 30.18 |
| | RACE | 3.76 | 17.62 | 16.95 | 31.58 | 5.70 | 9.32 | 14.16 | 7.843 | 30.08 |
| | SLD-Weak | 5.37 | 28.74 | 27.97 | 48.07 | 8.20 | 15.25 | 22.27 | 7.806 | 30.02 |
| | SLD-Medium | 4.19 | 25.29 | 12.71 | 34.39 | 7.60 | 10.17 | 15.73 | 10.435 | 29.48 |
| | SLD-Strong | 3.76 | 18.01 | 7.63 | 18.25 | 7.60 | 6.78 | 10.34 | 15.627 | 29.08 |
| | SLD-Max | 3.76 | 6.90 | 3.39 | 10.18 | 4.30 | 6.78 | 5.89 | 25.544 | 28.54 |
| | Blacklist | 6.02 | 13.79 | 26.27 | 17.54 | 6.20 | 15.25 | 14.18 | 7.298 | 30.06 |
| | POSI | 5.16 | 18.39 | 16.10 | 30.88 | 3.80 | 13.56 | 14.65 | 7.774 | 29.75 |
| | Ours | 3.19 | 10.98 | 11.02 | 2.92 | 4.23 | 8.47 | 6.80 | 7.365 | 30.10 |
| UCE | w/o defense | 11.49 | 44.06 | 73.73 | 37.89 | 39.30 | 35.59 | 40.34 | 4.379 | 31.02 |
| | RECE | 4.94 | 22.99 | 14.41 | 11.23 | 19.90 | 12.71 | 14.36 | 6.001 | 30.78 |
| | SLD-Weak | 12.57 | 50.57 | 67.80 | 42.11 | 45.00 | 36.44 | 42.42 | 5.635 | 30.87 |
| | SLD-Medium | 13.53 | 52.87 | 55.93 | 43.16 | 44.60 | 28.81 | 39.82 | 8.084 | 30.64 |
| | SLD-Strong | 13.10 | 44.83 | 45.76 | 38.95 | 42.70 | 27.12 | 35.41 | 12.412 | 30.31 |
| | SLD-Max | 12.03 | 37.93 | 33.90 | 26.67 | 35.50 | 20.34 | 27.73 | 18.445 | 29.90 |
| | Blacklist | 9.88 | 29.12 | 55.93 | 28.77 | 27.00 | 33.05 | 30.63 | 4.576 | 30.92 |
| | POSI | 9.13 | 27.20 | 30.51 | 22.81 | 10.60 | 26.27 | 21.09 | 5.270 | 30.56 |
| | Ours | 4.33 | 19.16 | 11.02 | 5.03 | 17.53 | 9.04 | 11.02 | 5.343 | 30.48 |

*Table 2.* Inference time of training-free defense methods.

| Defense | Inference Time (s) |
|---|---|
| w/o defense | 9.435 |
| Blacklist | 9.462 |
| SLD | 11.779 |
| Ours | 10.317 |

$\sigma_0$ and $k$, are detailed in Appendix D. The experiments are conducted using NVIDIA Tesla V100 GPUs.

**Baselines.** We compare our method's performance with various baselines. For pre-inference defense, we include commonly used adversarial erasing, RACE (Kim et al., 2024) for enhancing the robustness of ESD, and RECE (Gong et al., 2024) for UCE. Additionally, we consider inference-time safety mechanisms. These include input prompt modification methods, such as blacklist-based unsafe word removal (George, 2020) and input rewriting (POSI (Wu et al., 2024)). We also incorporate SLD (Schramowski et al., 2023) as a baseline representing approaches that alter the predicted noise direction during inference.

**Evaluation Metrics.** We concentrate on assessing robustness against adversarial prompts and the impact on the model's normal generation capabilities. For robustness evaluation, we consider adversarial prompts generated by various methods, including I2P (Schramowski et al., 2023), P4D (Chin et al., 2024), UDA (Zhang et al., 2024c), RAB (Tsai et al., 2024), MMA (Yang et al., 2024), and QF-PGD (Zhuang et al., 2023). Specific detectors (details in Appendix D) are used to identify the target concept in

the generated images. A lower detection rate indicates a lower attack success rate (ASR), thus demonstrating greater robustness. For evaluating model utility, we use the Fréchet Inception Distance (FID) (Heusel et al., 2017) for image quality and CLIP score (Hessel et al., 2021) for image-prompt alignment. The prompts for utility evaluation are sampled from the COCO validation set (Lin et al., 2014).

### 4.2. Comparisons with Baseline Methods

In this subsection, we employ various defenses to enhance the adversarial robustness of the unlearned models that have removed the concept of nudity. We compare our method with baseline approaches, emphasizing the trade-off between robustness and model utility, while also considering efficiency.

**Trade-Off Between Robustness and Utility.** The qualitative results demonstrating robustness are illustrated in Figure 3, while quantitative results are presented in Table 1. It is evident that, in the absence of defense mechanisms, the unlearned model is susceptible to various adversarial attacks. For the unlearned model processed by ESD, the SLD defense with maximum hyperparameters achieved the most robust performance, resulting in an average attack success rate (ASR) of 5.89%. However, this robustness comes at a significant cost to model utility, as indicated by the FID metric, which increased from 7.161 to 25.544. In contrast, our method achieves comparable robustness, with an average ASR of 6.80%, while effectively preserving the model's generative capabilities, reflected in an FID of 7.365. For the unlearned model processed by UCE, we observe that

Table 3. Ablation study of $\sigma_0$ and $k$.

| Unlearned Model | $\sigma_0$ | $k$ | ASR $\downarrow$ | FID $\downarrow$ | CLIP $\uparrow$ |
|---|---|---|---|---|---|
| ESD | 0.003 | | 12.54 | 7.252 | 30.16 |
| | 0.006 | 9 | 6.80 | 7.365 | 30.10 |
| | 0.009 | | 2.78 | 7.530 | 29.98 |
| | | 5 | 11.28 | 7.366 | 30.14 |
| | 0.006 | 9 | 6.80 | 7.365 | 30.10 |
| | | 13 | 4.98 | 7.370 | 30.02 |
| UCE | 0.006 | | 18.69 | 4.477 | 30.91 |
| | 0.009 | 9 | 14.46 | 4.766 | 30.77 |
| | 0.012 | | 11.02 | 5.343 | 30.48 |
| | | 5 | 17.71 | 5.237 | 30.54 |
| | 0.012 | 9 | 11.02 | 5.343 | 30.48 |
| | | 13 | 9.14 | 5.600 | 30.32 |

Table 4. Ablation study of global relevance.

| Relevance | ASR $\downarrow$ | FID $\downarrow$ | CLIP $\uparrow$ |
|---|---|---|---|
| Local | 15.03 | 4.800 | 30.72 |
| Local & Global | 11.02 | 5.343 | 30.48 |

Table 5. Ablation study of ViSU prompt pairs.

| Unlearned Model | Defense Method | ASR $\downarrow$ | FID $\downarrow$ | CLIP $\uparrow$ |
|---|---|---|---|---|
| ESD | w/o defense | 25.42 | 7.161 | 30.18 |
| | Ours (ViSU) | 6.80 | 7.365 | 30.10 |
| | Ours (Self-Gen) | 5.33 | 7.436 | 30.09 |
| UCE | w/o defense | 40.34 | 4.379 | 31.02 |
| | Ours (ViSU) | 11.02 | 5.343 | 30.48 |
| | Ours (Self-Gen) | 15.93 | 5.526 | 30.54 |

the SLD defense is less effective in enhancing adversarial robustness. In contrast, our method demonstrates superior defense performance, surpassing the adversarial erasing method (RECE), while also yielding better image quality, as indicated by the FID metric.

**Efficiency.** The adversarial erasing strategy necessitates additional training due to the iterative process of identifying adversarial inputs and erasing them by adjusting the parameters of the unlearned model. In contrast, the inference baselines do not require additional training, with the exception of POSI, which necessitates fine-tuning the LLM for prompt rewriting. To evaluate the efficiency of these training-free defense methods, we compare their inference time per image, and the results are presented in Table 2. Our method demonstrates superior inference efficiency compared to SLD. Although the simple blacklist-based defense is more efficient than ours, it is less effective in enhancing the robustness of unlearned models.

### 4.3. Ablation Study

In this subsection, we perform ablation studies to evaluate how various hyperparameters and components—including base noise level, noise scaling factor, global relevance incorporation, and prompt pair selection—affect the performance of our method.

**Base Noise $\sigma_0$ and Scaling Factor $k$.** The parameters $\sigma_0$ and $k$ are instrumental in controlling noise intensity. We apply different values of $\sigma_0$ and $k$ to both the ESD and UCE unlearned models and present the performance results in Table 3. While increasing $\sigma_0$ and $k$ enhances robustness, excessively large values may compromise model utility.

**Global Relevance.** We examine how performance differs when global context is incorporated into relevance computation, using the UCE unlearned model. The results are shown in Table 4. For vulnerable unlearned models, incorporating

global relevance further enhances robustness.

**ViSU Prompt Pairs.** In previous experiments, we used $7,863$ prompt pairs from the ViSU dataset (Poppi et al., 2024) to calculate the concept vector for the target concept ("*nudity*"), which is then utilized to compute the relevance score. To further demonstrate the effectiveness of our method, we implement nudity unlearning without using prompt pairs from ViSU. Specifically, we first use an uncensored LLM interface on Hugging Face to generate 100 nudity-related prompts (positive prompts) and then use another LLM interface (e.g., Kimi) to transform them into safe versions (i.e., negative prompts). These 100 self-generated prompt pairs (denoted as Self-Gen) are used in our experiments, with results shown in Table 5. Overall, our method remains effective even with only 100 collected prompt pairs.

### 4.4. Generalization Analysis

In this subsection, we evaluate the generalization ability of our method across different types of concepts and various T2I model variants.

**Generalization Across Concepts.** Previous experiments have demonstrated the effectiveness of our method in enhancing the robustness of unlearned models with the "*nudity*" concept erased. To further validate the generalizability of our approach, we evaluate it on additional concepts, including "*violence*" (NSFW), "*gun*" (object), and "*Van Gogh*" (style). Qualitative and quantitative results for "*violence*" are displayed in Figure 4 and Table 6, while results for "*gun*" and "*Van Gogh*" are provided in Appendix E.1. These results demonstrate that our method consistently achieves a superior trade-off between adversarial robustness and utility.

**Generalization Across T2I Model Variants.** While previous experiments were conducted on Stable Diffusion (SD) v1.4, we further evaluate our method on other widely used T2I models, SD v1.5 and SD v2.1, to assess its generalizability. The detailed results are presented in Appendix E.2.

*Table 6.* Quantitative results comparing different defense methods for unlearned models with the "*violence*" concept erased.

| Unlearned Model | Defense Method | Adversarial Robustness (Evaluated by ASR ↓) | | | | | Model Utility | |
|---|---|---|---|---|---|---|---|---|
| | | I2P | UDA | RAB | QF-PGD | Average | FID ↓ | CLIP ↑ |
| ESD | w/o defense | 17.59 | 38.33 | 50.00 | 28.33 | 33.56 | 7.656 | 29.75 |
| | RACE | 19.97 | 35.00 | 55.60 | 35.00 | 36.39 | 8.509 | 29.71 |
| | SLD-Weak | 16.01 | 32.50 | 42.27 | 29.17 | 29.98 | 8.392 | 29.67 |
| | SLD-Medium | 15.48 | 30.83 | 32.13 | 25.83 | 26.07 | 10.759 | 29.58 |
| | SLD-Strong | 12.57 | 27.50 | 21.07 | 20.83 | 20.49 | 14.477 | 29.43 |
| | SLD-Max | 9.39 | 16.67 | 9.47 | 17.50 | 13.26 | 19.274 | 29.23 |
| | Ours | 15.61 | 28.06 | 11.87 | 25.28 | 20.20 | 7.942 | 29.68 |
| UCE | w/o defense | 22.22 | 48.33 | 74.13 | 40.00 | 46.17 | 5.531 | 31.05 |
| | RECE | 18.92 | 39.17 | 60.40 | 31.67 | 37.54 | 6.743 | 30.90 |
| | SLD-Weak | 18.65 | 37.50 | 66.40 | 36.67 | 39.80 | 6.872 | 30.94 |
| | SLD-Medium | 18.78 | 35.00 | 55.07 | 30.83 | 34.92 | 9.327 | 30.49 |
| | SLD-Strong | 15.61 | 30.83 | 41.07 | 29.17 | 29.17 | 14.223 | 30.42 |
| | SLD-Max | 15.34 | 26.67 | 27.87 | 26.67 | 24.14 | 22.306 | 29.78 |
| | Ours | 18.21 | 30.00 | 11.29 | 29.72 | 22.31 | 6.063 | 30.56 |

## 5. Discussion

**Multiple Concept Unlearning.** Our method naturally extends to multi-concept unlearning scenarios. For instance, we conduct a preliminary exploration on the UCE unlearned model with "*nudity*" and "*violence*" concepts erased simultaneously. By computing token relevance scores as the maximum similarity value across multiple target concepts, our method effectively mitigates adversarial attacks while preserving model utility. Quantitative results are provided in Appendix F.1. These findings demonstrate the adaptability of our approach, though efficiency challenges may emerge when scaling to a large number of concepts.

**Comparison with Text Encoder Fine-tuning Methods.** Our method operates within a similar defense space as adversarial text encoder fine-tuning approaches, such as AdvUnlearn (Zhang et al., 2024b). However, key differences exist: AdvUnlearn requires computationally intensive fine-tuning and achieves empirical robustness, whereas our approach is training-free, applied at inference time, and provides both theoretical guarantees and empirical validation. A detailed discussion is presented in Appendix F.2.

**Failure Cases.** While our method generally demonstrates robust performance, we identify specific failure scenarios where certain adversarial prompts can circumvent our defense under the default hyperparameter $\sigma_0$. Increasing $\sigma_0$ can mitigate these cases but may also impair the model's performance on benign inputs. Future work should explore mechanisms to dynamically adjust $\sigma_0$ based on input characteristics, aiming to maintain robustness without compromising model utility. Detailed failure case analyses and qualitative examples are provided in Appendix F.3.

## 6. Conclusion

Improving the adversarial robustness of unlearned T2I models is important for content security. In this study, we derive the robust guarantee of generalized median smoothing with anisotropic noise, and propose an ***Adaptive Median Smoothing*** strategy with its efficient implementations for adversarial robustness enhancement of unlearned T2I diffusion models. Our inference-time defense method achieves a superior trade-off between adversarial robustness and model utility compared to previous methods.

## Acknowledgments

This work is supported by the National Natural Science Foundation of China (NSFC) under Grant No. 62372452. The authors would like to thank the anonymous reviewers for their insightful comments and constructive suggestions.

## Impact Statement

This work presents an inference-time defense strategy aimed at enhancing the adversarial robustness of unlearned text-to-image diffusion models. By reformulating robustness as a regression problem and introducing token-adaptive anisotropic noise, the proposed method achieves improved resilience against adversarial inputs while maintaining model utility and inference efficiency. These findings highlight a promising framework for securing generative models without compromising their core capabilities. We hope this work will inspire future research on adaptive defenses to safeguard generative models in increasingly sophisticated adversarial environments.

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

# A. Notation Summary

We present the key notations used throughout this paper in Table 7.

Table 7. Summary of key notations used in this paper.

| Notation | Description |
|---|---|
| $y$ | Benign input token embedding |
| $\mathcal{B}$ | Norm-bounded set defining allowed perturbations |
| $\delta$ | Adversarial perturbation constrained within $\mathcal{B}$ |
| $x_t$ | Noisy sample at timestep $t$ |
| $\epsilon^*(\cdot)$ | Robust noise prediction function |
| $\epsilon(\cdot)$ | Native (undefended) noise prediction function |
| $G$ | Isotropic Gaussian noise |
| $\mathcal{E}_p(\cdot)^i$ | Percentile-smoothed function for the $i$-th dimension under noise $G$ |
| $\Phi$ | Standard Gaussian cumulative distribution function |
| $G'$ | Anisotropic Gaussian noise |
| $\Sigma$ | Covariance matrix of anisotropic Gaussian noise (positive diagonal matrix) |
| $\mathscr{E}_p(\cdot)^i$ | Generalized percentile-smoothed function for the $i$-th dimension under noise $G'$ |
| $\|\delta\|_{\Sigma,2}$ | $\ell_2^{\Sigma}$-norm of $\delta$, defined as $\sqrt{\delta^{\top}\Sigma^{-1}\delta}$ |
| $\Delta_{\text{tgt}}$ | Target concept representation |
| $\Delta_{\text{glb}}^i$ | Global differential embedding for the $i$-th token |
| $\Delta_{\text{loc}}^i$ | Local differential embedding for the $i$-th token |
| $\text{sim}^i$ | Relevance of the $i$-th token to the target concept |
| $\sigma_0$ | Base noise magnitude |
| $k$ | Scaling factor controlling exponential growth of noise intensity |
| $h_{0.5}$ | Median text embedding served as the conditional input for the U-Net in diffusion models |

# B. Formulation Generalization for Subsequence Sampling

In practice, many text-to-image pipelines use DDIM (Song et al., 2021) or other fast samplers with $S \ll T$ steps (e.g., $S \approx 50$). Our original formulation covers the full DDPM (Ho et al., 2020) chain $(x_0, x_1, \ldots, x_T)$, which may be unnecessary for practical applications. To better align our theoretical framework with practical sampling efficiency, we generalize our formulation by considering a subsequence $\{x_{\tau_0}, x_{\tau_1}, \ldots, x_{\tau_S}\}$ used in practical sampling. These timesteps satisfy:

$$0 = \tau_0 < \tau_1 < \tau_2 < \cdots < \tau_S = T.$$

We then bound the KL divergence over this subsequence for any perturbation $\delta$ within the norm-bounded set $\mathcal{B}$:

$$\mathcal{D}_{\mathcal{KL}}\Big(p^*\big(x_{(\tau_0\ldots\tau_S)}|\mathcal{T}(y)\big) \,\|\, p^*\big(x_{(\tau_0\ldots\tau_S)}|\mathcal{T}(y+\delta)\big)\Big),$$

which translates to constraining the mean squared error (MSE) at each sampled step $\tau_i$:

$$\mathbb{E}_{x_{\tau_i},\tau_i}\big[\|\epsilon^*\big(x_{\tau_i},\mathcal{T}(y),\tau_i\big) - \epsilon^*\big(x_{\tau_i},\mathcal{T}(y+\delta),\tau_i\big)\|_2^2\big] \le C(\mathcal{B}) \quad \forall \delta \in \mathcal{B}.$$

# C. Proof of Theorem 3.2

**Lemma C.1.** *(Eiras et al., 2022) Let $f : \mathbb{R}^n \to [0,1]$ be a function, and define its smoothed version $g$ as follows:*

$$g(x) = \mathbb{E}_{G'\sim\mathcal{N}(0,\Sigma)}[f(x+G')],$$

*where $\Sigma$ is a positive diagonal matrix. Then, the function $\Phi^{-1}(g(x))$ is 1-Lipschitz with respect to the $\ell_2^{\Sigma}$ norm.*

**Theorem 3.2.** *The median-smoothed prediction with adversarial perturbation $\delta$ under anisotropic Gaussian noise $G' \sim \mathcal{N}(0,\Sigma)$ is bounded as follows:*

$$\underline{\mathscr{E}}_{\underline{p}'}(y)^i \le \mathscr{E}_{0.5}(y+\delta)^i \le \overline{\mathscr{E}}_{\overline{p}'}(y)^i \quad \forall\|\delta\|_{\Sigma,2} < \rho', \tag{7}$$

*where $\underline{p}' := \Phi\left(-\rho\prime\right)$ and $\overline{p}' := \Phi\left(\rho\prime\right)$ define the lower and upper probability bounds, respectively. $\|\delta\|_{\Sigma,2}$ denotes the $\ell_2^{\Sigma}$ norm of $\delta$, defined as $\|\delta\|_{\Sigma,2} = \sqrt{\delta^{\top}\Sigma^{-1}\delta}$.*

*Proof.* For clarity, we omit the superscript $i$ in our notation. Consider the event $\epsilon(y + \delta + G') \leq \overline{\mathscr{E}}_{\overline{p}'}(y)$ and let $\mathbb{1}_{\epsilon(y+\delta+G')\leq\overline{\mathscr{E}}_{\overline{p}'}(y)}$ denote the corresponding indicator function. We define $g(\delta)$ as the expectation of this indicator function:

$$g(\delta) = \mathbb{E}[\mathbb{1}_{\epsilon(y+\delta+G')\leq\overline{\mathscr{E}}_{\overline{p}'}(y)}] = \mathbb{P}[\epsilon(y + \delta + G') \leq \overline{\mathscr{E}}_{\overline{p}'}(y)].$$

Next, we consider the mapping defined by

$$\delta \mapsto \Phi^{-1}(g(\delta)).$$

According to the Lemma C.1, this mapping is 1-Lipschitz with respect to the $\ell_2^{\Sigma}$ norm. Therefore, we have:

$$\Phi^{-1}\big(\mathbb{P}[\epsilon(y + \delta + G) \leq \overline{\mathscr{E}}_{\overline{p}'}(y)]\big) - \Phi^{-1}\big(\mathbb{P}[\epsilon(y + G) \leq \overline{\mathscr{E}}_{\overline{p}'}(y)]\big) \geq -\|\delta\|_{\Sigma,2}.$$

Rearranging the above inequality yields:

$$\begin{aligned}
\Phi^{-1}\big(\mathbb{P}[\epsilon(y + \delta + G) \leq \overline{\mathscr{E}}_{\overline{p}'}(y)]\big) &\geq \Phi^{-1}\big(\mathbb{P}[\epsilon(y + G) \leq \overline{\mathscr{E}}_{\overline{p}'}(y)]\big) - \|\delta\|_{\Sigma,2} \\
&\geq \Phi^{-1}\big(\mathbb{P}[\epsilon(y + G) \leq \overline{\mathscr{E}}_{\overline{p}'}(y)]\big) - \rho' \\
&\geq \Phi^{-1}(\overline{p}') - \rho' \\
&= \Phi^{-1}(\Phi(\rho')) - \rho' \\
&= 0.
\end{aligned}$$

Since $\Phi^{-1}(0.5) = 0$ and the function $\Phi^{-1}(\cdot)$ is monotonically increasing, it follows that:

$$\mathbb{P}[\epsilon(y + \delta + G) \leq \overline{\mathscr{E}}_{\overline{p}'}(y)] \geq 0.5.$$

According to the definition of $\overline{\mathscr{E}}_{0.5}(\cdot)$, we have:

$$\overline{\mathscr{E}}_{0.5}(y + \delta) = \inf\{v \in \mathbb{R} \mid \mathbb{P}[\epsilon(y + \delta + G) \leq v] \geq 0.5\}.$$

Thus, we can conclude that:

$$\overline{\mathscr{E}}_{0.5}(y + \delta) \leq \overline{\mathscr{E}}_{\overline{p}'}(y).$$

Similarly, we can derive the inequality on the other side:

$$\underline{\mathscr{E}}_{0.5}(y + \delta) \geq \underline{\mathscr{E}}_{\underline{p}'}(y).$$

Recall that we denote $\mathscr{E}_{0.5}$ as the percentile-smoothed function applicable to either definition. This notation allows us to represent both $\underline{\mathscr{E}}_{0.5}$ and $\overline{\mathscr{E}}_{0.5}$ simultaneously. With established bounds for both functions, we can express our result concisely:

$$\underline{\mathscr{E}}_{\underline{p}'}(y) \leq \mathscr{E}_{0.5}(y + \delta) \leq \overline{\mathscr{E}}_{\overline{p}'}(y) \quad \forall \|\delta\|_{\Sigma,2} < \rho'.$$

This completes the proof.

# D. Implementation Details

**Hyperparameters.** The number of Monte Carlo samples, denoted as $n$, is set to 13. For the ESD unlearned model following the erasure of the nudity concept, we set $\sigma_0$ to 0.006 and $k$ to 9, utilizing only local relevance scores. In the case of the UCE unlearned model after erasing the nudity concept, $\sigma_0$ is set to 0.012 and $k$ to 9, incorporating both local and global relevance scores. For the ESD unlearned model after erasing the violence concept, $\sigma_0$ remains at 0.006 and $k$ at 9, again using only local relevance scores. Finally, for the UCE unlearned model after erasing the violence concept, $\sigma_0$ is set to 0.012 and $k$ to 9, utilizing only local relevance scores.

**Unsafe Content Detectors.** We employ Nudenet (Bedapudi, 2019) for nudity content detection, identifying nine specific exposed body areas: buttocks, female breasts, female genitalia, anus, male genitalia, male breasts, belly, feet, and armpits. The detection uses a confidence threshold of 0.6. For violence content detection, we utilize the fine-tuned Q16 model provided by (Qu et al., 2023).

# E. Additional Generalization Analysis

This section presents additional generalization results across various concepts and text-to-image (T2I) model variants.

### E.1. Generalization Across Concepts

For the object concept, we target "*gun*" and use three attack methods (UDA, RAB, QF-PGD) to assess adversarial robustness, reporting the mean attack success rate (ASR). We compare our approach with SLD baselines. The results, presented in Table 8, show that our method achieves a superior balance between robustness and utility. For the style concept, we target "*Van Gogh*" using the same three attack methods, with results shown in Table 9. These results demonstrate our method's effectiveness in addressing both object and style concepts.

*Table 8.* Quantitative results comparing different defense methods for unlearned models with the "*gun*" concept erased.

| Unlearned Model | Defense Method | ASR ↓ | FID ↓ | CLIP ↑ |
|---|---|---|---|---|
| ESD | w/o defense | 56.11 | 6.595 | 30.06 |
| | SLD-Weak | 46.78 | 7.521 | 29.90 |
| | SLD-Medium | 28.44 | 9.624 | 29.45 |
| | SLD-Strong | 22.33 | 12.510 | 28.88 |
| | SLD-Max | 18.11 | 16.630 | 28.26 |
| | Ours | 26.07 | 7.959 | 29.66 |
| UCE | w/o defense | 58.56 | 5.281 | 31.03 |
| | SLD-Weak | 45.44 | 6.133 | 31.02 |
| | SLD-Medium | 40.11 | 8.201 | 30.65 |
| | SLD-Strong | 32.11 | 10.582 | 30.12 |
| | SLD-Max | 21.00 | 14.223 | 29.25 |
| | Ours | 29.85 | 7.142 | 30.29 |

*Table 9.* Quantitative results comparing different defense methods for unlearned models with the "*Van Gogh*" concept erased.

| Unlearned Model | Defense Method | ASR ↓ | FID ↓ | CLIP ↑ |
|---|---|---|---|---|
| ESD | w/o defense | 11.33 | 5.612 | 30.38 |
| | SLD-Weak | 6.00 | 6.133 | 30.37 |
| | SLD-Medium | 1.33 | 7.929 | 30.03 |
| | SLD-Strong | 0.00 | 10.971 | 29.46 |
| | SLD-Max | 0.00 | 15.364 | 28.70 |
| | Ours | 0.00 | 6.181 | 30.16 |
| UCE | w/o defense | 39.83 | 2.024 | 31.11 |
| | SLD-Weak | 24.50 | 4.035 | 30.98 |
| | SLD-Medium | 1.67 | 6.569 | 30.55 |
| | SLD-Strong | 0.00 | 10.692 | 29.56 |
| | SLD-Max | 0.00 | 17.322 | 28.30 |
| | Ours | 0.00 | 3.843 | 30.94 |

## E.2. Generalization Across T2I Model Variants

We use the UCE unlearned model with"*nudity*" removed, keeping $\sigma_0$ and $k$ consistent with version 1.4. Table 10 shows results using four attack methods: I2P, P4D, RAB, and QF-PGD, with average attack success rate (ASR) for robustness and FID and CLIP scores for utility. Our findings indicate that our method remains effective across different T2I model variants, enhancing the adversarial robustness of unlearned models without significantly affecting their generation capabilities.

*Table 10.* Comparison of our method and baselines on UCE-unlearned models across different SD versions.

| SD Version | Defense Method | ASR ↓ | FID ↓ | CLIP ↑ |
|---|---|---|---|---|
|  | w/o defense | 28.05 | 4.354 | 31.05 |
|  | RECE | 9.02 | 7.589 | 30.52 |
| 1.5 | Blacklist | 24.49 | 4.557 | 30.94 |
|  | POSI | 20.13 | 5.243 | 30.23 |
|  | Ours | 10.65 | 5.460 | 30.56 |
|  | w/o defense | 21.52 | 5.303 | 31.00 |
|  | RECE | 2.08 | 11.856 | 29.33 |
| 2.1 | Blacklist | 14.04 | 5.511 | 30.91 |
|  | POSI | 17.77 | 6.175 | 30.15 |
|  | Ours | 7.01 | 6.575 | 30.66 |

# F. Detailed Discussion

In this section, we provide a detailed discussion from three perspectives: multiple concept unlearning, comparative analysis with text encoder fine-tuning methods (AdvUnlearn), and failure cases.

## F.1. Multiple Concept Unlearning

For this multi-concept setting, we compute each token's relevance score by taking the maximum of its similarity with both the nudity concept direction and the violence concept direction. The results are presented in Table 11, where for each concept, the average attack success rate (ASR) is computed across three attack methods (I2P, RAB, and QF-PGD). Our method is effective in concurrently handling multiple concepts. Compared to the single-concept case, the additional computation involves calculating similarities with more concepts and performing a subsequent max operation. However, when the number of concepts is large, the efficiency of our method may be impacted.

*Table 11.* Performance of our defense method on a multi-concept unlearned model.

| Defense | ASR (Nudity) ↓ | ASR (Violence) ↓ | FID ↓ | CLIP ↑ |
|---|---|---|---|---|
| w/o defense | 24.22 | 36.30 | 6.181 | 30.92 |
| Ours | 5.26 | 9.18 | 6.752 | 30.31 |

## F.2. Comparative Analysis with AdvUnlearn

We provide an extended comparative analysis between our approach and AdvUnlearn (Zhang et al., 2024b), focusing on the following three aspects.

**Methodological Differences.** AdvUnlearn is a pre-inference method that fine-tunes the text encoder. It falls into the category of adversarial erasing-based defenses, similar to the RACE (Kim et al., 2024) and RECE (Gong et al., 2024) baselines evaluated in our paper. While AdvUnlearn operates in the text-embedding space as our method does, several key differences exist:

- **Efficiency**: AdvUnlearn requires relatively large computational resources for the fine-tuning process, as acknowledged by the authors (Zhang et al., 2024b). In contrast, our approach is training-free and operates at inference time, offering greater efficiency.
- **Theoretical Guarantees**: Unlike AdvUnlearn's empirical defense, we provide theoretical guarantees through generalized median smoothing (Theorem 3.2), which could potentially offer new insights to this field.

**Comparative Performance Evaluation.** We implement AdvUnlearn using its official codebase to compare it with our method, targeting the concept of "*nudity*". For our approach, we integrate our method with ESD, setting $\sigma_0$ to 0.012 and $k$ to 9. We evaluate adversarial robustness through four attacks (i.e., I2P, RAB, MMA, and QF-PGD) and calculate the average attack success rate (ASR), while assessing model utility through FID and CLIP scores. The results, presented in Table 12, show that our method achieves adversarial robustness comparable to AdvUnlearn. In terms of model utility, AdvUnlearn maintains superior FID metrics (image quality) because it preserves the U-Net weights. However, its CLIP score (image-text alignment) is lower due to modifications in the text encoder. Our defense employs adaptive median smoothing without altering text encoder parameters, better preserving image-text alignment capabilities.

**Compatibility and Future Work.** We find that directly applying our method to an unlearned text encoder presents challenges. Our approach requires calculating each token's relevance to target concepts, but AdvUnlearn's fine-tuning process maps target token representations to benign ones, making it difficult to distinguish between them based on textual representations. This results in inaccurate relevance scores. Currently, our proposed method serves as an effective complement to unlearning approaches that modify U-Net parameters, which constitute the majority of diffusion model unlearning techniques (Gandikota et al., 2023; Kumari et al., 2023; Gandikota et al., 2024). In future work, we plan to explore adaptations to our method to enhance compatibility with unlearned text encoders.

*Table 12.* Comparison of AdvUnlearn and our defense on the "*nudity*" concept.

| **Defense** | **ASR $\downarrow$** | **FID $\downarrow$** | **CLIP $\uparrow$** |
|---|---|---|---|
| AdvUnlearn | 1.39 | 6.973 | 29.03 |
| Ours | 1.62 | 7.783 | 29.80 |

### F.3. Failure Cases

We provide some qualitative results of failure cases in Figure 5. These results show that with the default hyperparameter value of $\sigma_0$ (as seen in the second and fifth columns), certain adversarial prompts can still restore the nudity concept. In our implementation, $\sigma_0$ is fixed for each unlearned model, but there are challenging cases that require a larger $\sigma_0$ to address effectively. While increasing $\sigma_0$ can help defend against these prompts, it may also degrade model performance with benign inputs. Therefore, it is worthwhile to explore how to dynamically adjust $\sigma_0$ based on the input prompt in future work.

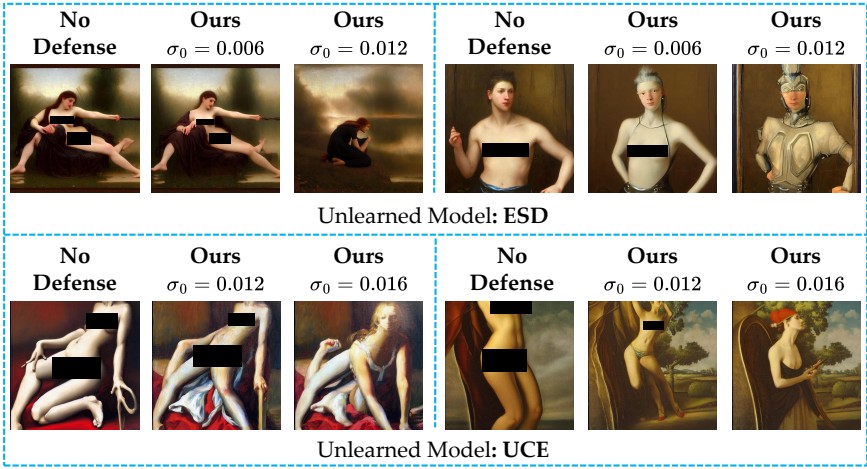

*Figure 5.* Qualitative results of failure cases. The images illustrate scenarios where adversarial prompts can bypass the default defense settings, partially restoring the unlearned concept (columns 2 & 5). Increasing $\sigma_0$ (columns 3 & 6) improves robustness but may impact benign-generation quality, highlighting a trade-off between robustness and utility.

