# OpenReview forum: "Adaptive Median Smoothing: Adversarial Defense for Unlearned Text-to-Image Diffusion Models at Inference Time"
_ICML.cc/2025/Conference — ICML 2025 poster_

### Official Review · Reviewer_FnEb · 2025-02-23

**Overall Recommendation:** 4

**Summary:**

This paper seeks to enhance the adversarial robustness of unlearned t2i diffusion models, specifically aiming to balance the adversarial robustness and generative capabilities of the original t2i models. The proposed method, Adaptive Median Smoothing, starts by formulating the target task as a regression problem, and extends it to an anisotropic noise case with median smoothing by introducing a global relevance score for the input prompt.

**Claims And Evidence:**

The main claims of the paper are supported by the experimental results and ablation studies.

**Essential References Not Discussed:**

The paper has a relative good reference discussion on related works in the problem of adversarial robustness of unlearned models.

**Experimental Designs Or Analyses:**

The current experimental designs and analyses are in general reasonable, with several questions specified below.

**Methods And Evaluation Criteria:**

The proposed method, with regression formulation for unlearned diffusion models, generally makes sense to the reviewer (with some more specific questions detailed in the Questions below). The evaluation results are reported based on three metrics, including ASR, FID, and CLIPScore, which also makes sense.

**Other Comments Or Suggestions:**

- The abstract is a bit too long, maybe the authors may consider shortening it for better readability.

- The reviewer believes it will be interesting to investigate a bit further the combination of different unlearned concepts, as those concepts may present correlations among them and impact the performance of the proposed method.

- Maybe also consider numbering all the equations as it is difficult to refer to while some are numbered and others are not.

**Other Strengths And Weaknesses:**

S1: The paper is well-structured with a clear logical flow, making it easy to follow.

S2: The problem and task are relatively well-defined and appear reasonable.

W1: This work focuses on a highly specific scenario—adversarial robustness for unlearned T2I diffusion models—with experiments conducted only on SD 1.4. It remains unclear how well the proposed method generalizes to other T2I model variants and how its performance is affected by the underlying unlearned base models.

W2: The experiments primarily evaluate two concepts: nudity and violence. While I am not explicitly requesting additional experiments, I would appreciate a discussion on specific failure cases to better understand the limitations of the approach.

W3: Certain aspects of the formulation and implementation details remain unclear—please see my questions section for further clarification.

**Questions For Authors:**

Q1: In the regression formulation, are there any constraints on the expected perturbation $\delta$? Specifically, could the perturbation become too large, such that $y+\delta$ falls into a distribution different from  $p*(x_{(0,...,T)|\mathcal{T}(y)})$ ?

Q2: Why does the formulation consider all intermediate steps in the regression process? Intuitively, wouldn’t it make more sense to focus only on step 0 in theory? Moreover, in implementation, many prior works suggest that the full sequence is not necessary for T2I generation.

Q3: Regarding relevance score computation, the paper mentions the need to collect pairs of positive and negative prompts. How are these pairs collected and constructed, and what is the computational cost associated with this step?

Q4: Do the authors have any insights on failure cases? Additionally, how might the method be impacted if multiple concepts were unlearned in a single base T2I model?

**Relation To Broader Scientific Literature:**

The reviewer finds that the references and related literature in the paper are generally well discussed, helping the reader understand the key contributions proposed in this work.

**Theoretical Claims:**

The main theoretical claim is Theorem 3.2 in P4, which is just a special variant of Lemma 1 under the constraint of non-isotropic Gaussian noise, the reviewer checked the proof and did not find evident flaws.

---

> ### Author Rebuttal · Authors · 2025-04-01
>
> # Generalizability Across T2I Model Variants
> We evaluated our method on other widely used T2I models, SD 1.5 and SD 2.1, to assess the generalizability of our method. Due to time constraints, we used the UCE unlearned model with "*nudity*" removed, keeping $\sigma_0$ and $k$ consistent with version 1.4. The table below shows results using four attack methods: I2P, P4D, RAB, and QF-PGD, with average attack success rate (ASR) for robustness and FID and CLIP scores for utility.
> | SD Version | Defense | ASR $\downarrow$ | FID $\downarrow$ | CLIP $\uparrow$ |
> | -------- | --------- | ----------- | ----------- | --------------- |
> | 1.5| w/o defense|28.05|4.354|31.05|
> | 1.5|Ours|10.65|5.460|30.56|
> | 2.1|w/o defense|21.52|5.303|31.00|
> | 2.1|Ours|7.01|6.575|30.66|
>
> Our findings indicate that our method remains effective across different T2I model variants, enhancing the adversarial robustness of unlearned models without significantly affecting their generation capabilities. We will expand the analysis to include additional baselines in our paper.
> # Discussion on Failure Cases
> Thank you for your insightful feedback. We have provided some qualitative results [here](https://anonymous.4open.science/r/Re-ML/fig5.png). These results show that with the default hyper-parameter value of $\sigma_0$ (as seen in the second and fifth columns), certain adversarial prompts can still restore the nudity concept. In our implementation, $\sigma_0$ is fixed for each unlearned model, but there are challenging cases that require a larger $\sigma_0$ to address effectively. While increasing $\sigma_0$ can help defend against these prompts, it may also degrade model performance with benign inputs. Therefore, it is worthwhile to explore how to dynamically adjust $\sigma_0$ based on the input prompt in future work. We will include this discussion in our paper.
>
> # Multiple Concept Unlearning
> Thank you for the insightful comment. We conducted a preliminary exploration on the UCE unlearned model with *nudity* and *violence* concepts erased simultaneously. For this multi-concept setting, we computed each token's relevance score by taking the maximum of its similarity with both the nudity concept direction and the violence concept direction. The results are presented in the table below, where for each concept, the average attack success rate (ASR) is computed across three attack methods (I2P, RAB, and QF-PGD).
> | Defense     | ASR $\downarrow$ (Nudity) | ASR $\downarrow$ (Violence) | FID $\downarrow$ | CLIP $\uparrow$ |
> | ----------- | ------------------------- | ----------------------- | ------------ | ----------- |
> | w/o defense|24.22|36.30|6.181|30.92|
> | Ours|5.26|9.18| 6.752|30.31|
>
> Our method is effective in concurrently handling multiple concepts. Compared to the single-concept case, the additional computation involves calculating similarities with more concepts and performing a subsequent max operation. However, when the number of concepts is large, the efficiency of our method may be affected. We will include this analysis in our paper.
> # Constraints on Adversarial Perturbation
> The perturbation is constrained within a norm-bounded set $\\mathcal{B}$, specifically $\||\delta\||\_2<\rho$ in the isotropic case (Lemma 3.1) and $\||\delta\||_{\Sigma,2}<\rho\prime$ in the anisotropic case (Theorem 3.2). If the upper bound of the perturbation norm is large, users can increase noise intensity in median smoothing. A detailed discussion on handling large perturbations is available in the `Theoretical Tightness under Large Perturbation` section of our response to Reviewer CcZX.
>
> # Justification for Considering Intermediate Steps in Formulation
> Our formulation follows the DDPM [1] framework, modeling generation as Markov chains. While we aim to constrain $x_0$, the dependency structure ($x_0$ depends on $x_1$, $x_1$ on $x_2$, and so forth) means perturbations propagate through the entire sequence. The formulation ensures robustness at each step, preventing cascading errors and ultimately safeguarding the final output. Notably, related works, such as [2], also incorporate intermediate steps in their formulations.
>
> In practical applications, efficient sampling techniques enable us to perform smoothing over a limited number of time steps, often just tens or even fewer, thus maintaining computational efficiency.
>
> [1] Denoising Diffusion Probabilistic Models, NeurIPS 2020.
>
> [2] Ablating Concepts in Text-to-Image Diffusion Models, ICCV 2023.
>
> # Collection of Prompt Pairs
> Positive and negative prompts are sourced from the ViSU dataset, as explained in the Implementation Details (Section 4.1), requiring no additional computation. We also explored generating prompt pairs using a large language model (LLM) interface. More details are in the `Clarifying the Use of Prompt Pairs from ViSU` section of our response to Reviewer 9A3r.
>
> # Additional Comments
> We will shorten the abstract and number all the equations in our paper. Thanks for your advice.

---

> > ### Comment · Reviewer_FnEb · 2025-04-04
> >
> > (Accidentally posted this Rebuttal Comment as Official Comment before)
> >
> > First, I thank the authors for their rebuttal. I have read the responses, and most of my concerns have been addressed. Accordingly, I am raising my score to a 4.
> >
> > That said, I remain unconvinced by the justification regarding the use of all intermediate steps. In most T2I diffusion models, DDIM-based samplers with significantly reduced steps (e.g., ~50 steps) are commonly adopted. In this context, the original DDPM sampling procedure is often considered unnecessary. Given the application-oriented nature of this work, I do not see a strong rationale for adhering to the theoretically bounded stepwise error formulation. While the authors mention that “efficient sampling techniques enable us to perform smoothing over a limited number of time steps,” it appears that no concrete experiments have been conducted.

---

> > > ### Author Response · Authors · 2025-04-06
> > >
> > > We sincerely appreciate the continued engagement and your raising of our score.
> > >
> > > You are correct that in practice many T2I pipelines use DDIM [1] or other fast samplers with $S \ll T$ steps (e.g., $S \approx 50$). Our original formulation covers the full DDPM [2] chain ($x_0,x_1,\dots, x_T$), which may not be necessary in practical applications. To better align our theoretical framework with practical sampling procedures, we generalize our formulation by considering a **sub-sequence** $\\{x_{\tau_0},x_{\tau_1},\dots, x_{\tau_S}\\}$ used in **practical sampling**. These timesteps satisfy:
> > > $$
> > > 0 = \tau_0 < \tau_1 < \tau_2 < \dots < \tau_S = T.
> > > $$
> > > We then bound the KL divergence over this **sub-sequence** for any perturbation $\delta$ within a norm-bounded set $\mathcal{B}$:
> > >
> > > $$
> > > \mathcal{D}_{\mathcal{KL}}\Big({p^*}\big({x}\_{(\tau_0\dots\tau_S)}|\mathcal{T}(y)\big)\\,||\\,{p^*}\big({x}\_{(\tau_0\dots\tau_S)}|\mathcal{T}(y+\delta)\big)\Big),
> > > $$
> > >
> > > which translates to constraining the mean squared error (MSE) at **each sampled step** $\tau_i$, for $i=1,\dots,S$:
> > >
> > > $$
> > > \mathbb{E}_{{x}\_{\tau_i},\tau_i}\big[\\|{\epsilon}^{\*}\big({x}\_{\tau_i},\mathcal{T}(y),\tau_i\big)-{\epsilon}^{\*}\big({x}\_{\tau_i},\mathcal{T}(y+\delta),\tau_i\big)\\|_2^2\big] \leq C(\mathcal{B}), \quad \forall \delta \in \mathcal{B}.
> > > $$
> > >
> > > In our **experiments** we use $S=50$ for SD 1.4/1.5 and $S=25$ for SD 2.1, applying adaptive median smoothing at **each sampled step**. We will clarify this generalized formulation in the final version.
> > >
> > > Thank you again for helping us enhance the rigor and clarity of our paper.
> > >
> > > [1] Denoising Diffusion Implicit Models, ICLR 2021.
> > >
> > > [2] Denoising Diffusion Probabilistic Models, NeurIPS 2020.

---

### Official Review · Reviewer_ZpJv · 2025-03-01

**Overall Recommendation:** 4

**Summary:**

This paper proposes an inference-time defense method, named Adaptive Median Smoothing, to protect unlearned text-to-image diffusion models against adversarial prompt attacks. Specially, the promposed method reformulates robustness as a regression problem and extends median smoothing by using anisotropic noise. Then, it utilizes token-level adaptive noise to keep the model robust without hurting image generation utility.

**Claims And Evidence:**

In this paper, authors make five claims about their proposed method
-  Robustness enhancement is supported by its lower ASR (through different attack methods)
- Model utility preservation is supported by the results of FID and CLIP score, which assess the image generation quality and text alignment, respectively.
- Compared with other training-free methods, it utilizes reasonable inference time for a robust defense.
- Good generalization across different NSFW contents (e.g., nudity, violence).
- Supported by Theorem 3.2 and extensive experiments, the claim that the proposed method improve robustness against adversarial prompts while maintaining the model utility generative quality and fast inference.

**Essential References Not Discussed:**

NA

**Experimental Designs Or Analyses:**

A SOTA adversarial unlearning[1] is a strong baseline because, unlike ESD and UCE, it fine-tunes the text encoder to achieve effective unlearning and also maintain good model utility, which exist certain resemblances to the proposed method in essence since both are implementing defense in the text embedding space. It is also worth investigating whether the proposed defense can further enhance unlearning performance for unlearned text encoder.

[1] Defensive Unlearning with Adversarial Training for Robust Concept Erasure in Diffusion Models, NeurIPS 2024

**Methods And Evaluation Criteria:**

The proposed method utilizes a new form of median smoothing with anisotropic noise and adapts noise intensity per token using concept relevance scores.

**Other Comments Or Suggestions:**

NA

**Other Strengths And Weaknesses:**

NA

**Questions For Authors:**

NA

**Relation To Broader Scientific Literature:**

NA

**Theoretical Claims:**

Theorem 3.2 provides bounds for robust regression using median smoothing with anisotropic noise.

---

> ### Author Rebuttal · Authors · 2025-04-01
>
> Thank you for your insightful comment regarding the relationship between our method and AdvUnlearn [1]. Our response is elaborated on the following three aspects:
>
> # Methodological Differences
>
> AdvUnlearn [1] is a pre-inference method that fine-tunes the text encoder. It falls into the category of **adversarial erasing-based** defenses, similar to the *RACE* and *RECE* baselines evaluated in our manuscript. While AdvUnlearn operates in the text-embedding space as our method does, several key differences exist:
>
> -  **Efficiency**: AdvUnlearn requires relatively large computational resources for the fine-tuning process, as acknowledged by the authors [1]. In contrast, our approach is training-free and operates at inference time, offering greater efficiency.
> - **Theoretical Guarantees**: Unlike AdvUnlearn’s empirical defense, we provide theoretical guarantees through generalized median smoothing (Theorem 3.2), which could potentially offer new insights to this field.
>
> # Comparative Performance Evaluation
>
> We implemented AdvUnlearn using its official codebase to compare it with our method, targeting the concept of "*nudity*". For our approach, we integrated our method with ESD, setting $\sigma_0$ to 0.012 and $k$ to 9. We evaluated adversarial robustness through four attacks (i.e., I2P, RAB, MMA, and QF-PGD) and calculated the average attack success rate (ASR), while assessing model utility through FID and CLIP scores. The results are shown in the table below:
>
> | Defense    | ASR $\downarrow$ | FID $\downarrow$ | CLIP $\uparrow$ |
> | ---------- | ---------------- | ---------------- | --------------- |
> | AdvUnlearn | 1.39             | 6.973            | 29.03           |
> | Ours       | 1.62             | 7.783            | 29.80           |
>
> The results indicate that our method achieves adversarial robustness comparable to AdvUnlearn. In terms of model utility, AdvUnlearn maintains superior FID metrics (image quality) because it preserves the U-Net weights. However, its CLIP score (image-text alignment) is lower due to modifications in the text encoder. Our defense employs adaptive median smoothing without altering text encoder parameters, better preserving **image-text alignment** capabilities.
>
> # Compatibility and Future Work
>
> We found that directly applying our method to an unlearned text encoder presents challenges. Our approach requires calculating each token's relevance to unsafe concepts, but AdvUnlearn's fine-tuning process maps unsafe token representations to benign ones, making it difficult to distinguish between them based on textual representations. This results in inaccurate relevance scores.
>
> Currently, our proposed method serves as an effective complement to unlearning approaches that modify *U-Net parameters*, which constitute the **majority** of diffusion model unlearning techniques. In future work, we plan to explore adaptations to our method to enhance compatibility with unlearned text encoders.
>
> We will incorporate a citation to AdvUnlearn and include the above analysis in our paper.
>
> [1] Defensive Unlearning with Adversarial Training for Robust Concept Erasure in Diffusion Models, NeurIPS 2024.

---

### Official Review · Reviewer_CcZX · 2025-03-14

**Overall Recommendation:** 2

**Summary:**

This paper proposes "Adaptive Median Smoothing" as an inference-time defense for adversarial attacks on unlearned diffusion models. The defense goal can be formulated as minimizing MSE predicted noise before and after adversarial perturbation. Based on this formulation, the paper then introduces the naive median smoothing method with isotropic noise, analyzing its bound to show the method can theoretically defend against adversarial attacks. The paper then generalizes the analysis to generalized median smoothing with anisotropic noise. However, such an approach is computationally heavy in approximating medians for each timestep. Given these insights, the paper proposes adaptive median smoothing: (1) it computes the relevance score of each token by looking into the text embedding, and adds larger noise to the token embeddings of more relevant tokens; (2) instead of computing medians independently at every timestep, the paper computes medians of token embeddings before the generation and always use this as conditional input to the denoiser. The paper conducts defense experiments on ESD and UCE under different adversarial attacks, focusing on the concept nudity. It compares the ASR, FID, and CLIP with baseline defense methods and shows better balance between adversarial robustness and utility.

**Claims And Evidence:**

1. Why the estimation of medians before generation can still be theoretically supported is not clear - the theoretical results seem to assume the medians are estimated separately for every timestep.
2. The bound of mean squared error requires $\|\delta\|_2<\rho$, yet the norm upper bound of adversarial perturbation $\delta$ can be very large. It's not clear how meaningful the bound of the mean squared error in equation (2) is - if it's a loose bound, it can not theoretically support that median smoothing can be used as a defense strategy.

**Essential References Not Discussed:**

[1] Defensive Unlearning with Adversarial Training for Robust Concept Erasure in Diffusion Models. Yimeng Zhang, etc. NeurIPS 2024.
Ths paper comes up with a similar regression formulation in defending adversarial perturbations and maybe cited and discussed.

**Experimental Designs Or Analyses:**

1. There are concerns about datasets, as stated in the Evaluation Criteria section.
2. Besides, the paper should discuss why only UCE and ESD are the only chosen unlearned models - is it because they are the most robust and widely used models, or is it for some other reasons?

**Methods And Evaluation Criteria:**

The method is for defending unlearned diffusion models from adversarial attacks during inference-time. Yet, the benchmark datasets only focus on defending nudity. In machine unlearning, usually, at least three levels of concepts are considered: object, style, NSFW. Since the key problem the paper aims to solve is defending unlearned diffusion models, it is essential to fully demonstrate its effectiveness on these different levels of concepts, instead of only nudity from NSFW.

**Other Comments Or Suggestions:**

It might be helpful to have a notation section - not a big problem.

**Other Strengths And Weaknesses:**

Strengths
1. The paper does stand out with its theoretical analysis on applying median smoothing to defending unlearned diffusion models.
2. The paper does make an interesting engineering effort to adapt naive median smoothing to a more efficient defending method with good performance on defending nudity.

**Questions For Authors:**

1. Why the estimation of medians before generation can still be theoretically supported?
2. When assuming $\|\delta\|_2<\rho$, is it possible that $\rho$ can be very large and cause the bound of the mean squared error in equation (2) to be a loose bound? How to argue it is a meaningful bound?
3. Can the method also work well in defending concepts in the category of object and style?
4. Why only UCE and ESD are the only chosen unlearned models - is it because they are the most robust and widely used models, or is it for some other reasons?

**Relation To Broader Scientific Literature:**

Maybe the paper can inspire later work to consider applying methods inspired from signal processing to the deep learning safety community for a more fundamental and potentially fruitful research.

**Theoretical Claims:**

I have checked the correctness of proofs, and it overall makes sense following the assumptions.

---

> ### Author Rebuttal · Authors · 2025-04-01
>
> # Evaluation Across Object and Style Concepts
> We would like to clarify that our original manuscript includes experiments not only on the concept of nudity but also on **violence**. Following your suggestion, we expanded to include **object** and **style** concepts.
> - For the **object** concept, we targeted "*gun*" and used three attack methods (UDA, RAB, QF-PGD) to assess adversarial robustness, reporting the mean attack success rate (ASR). Due to time constraints, we compared only with SLD-Medium and SLD-Max baselines. The results, presented in the table below, show our method achieves a superior balance between robustness and utility.
> | Unlearned Model | Defense     | ASR $\downarrow$ | FID $\downarrow$ | CLIP $\uparrow$ |
> | --------------- | ----------- | ---------------- | ---------------- | --------------- |
> | ESD | w/o defense | 56.11  | 6.595            | 30.06   |
> | ESD | SLD-Medium  | 28.44 | 9.624            | 29.45  |
> | ESD | SLD-Max     | 18.11 | 16.630           | 28.26  |
> | ESD | Ours        | 26.07 | 7.959            | 29.66  |
> | UCE | w/o defense | 58.56 | 5.281    | 31.03           |
> | UCE | SLD-Medium  | 40.11 | 8.201            | 30.65           |
> | UCE | SLD-Max | 21.00 | 14.223           | 29.25           |
> | UCE | Ours        | 29.85 | 7.142            | 30.29           |
>
> - For the **style** concept, we targeted "*Van Gogh*" using three attack methods (UDA, RAB, QF-PGD), with results shown in the table below. These results demonstrate the method's effectiveness in addressing style concepts.
> | Unlearned Model | Defense     | ASR $\downarrow$ | FID $\downarrow$ | CLIP $\uparrow$ |
> | --------------- | ----------- | ---------------- | ---------------- | --------------- |
> | ESD | w/o defense | 11.33     | 5.612 | 30.38 |
> | ESD | SLD-Medium  | 1.33    | 7.929 | 30.03 |
> | ESD | SLD-Max     | 0.00     | 15.364  | 28.70 |
> | ESD | Ours        | 0.00       | 6.181   | 30.16  |
> | UCE | w/o defense | 39.83   | 2.024    | 31.11   |
> | UCE | SLD-Medium  | 1.67    | 6.569    | 30.55  |
> | UCE | SLD-Max     | 0.00     | 17.322    | 28.30  |
> | UCE | Ours        | 0.00     | 3.843    | 30.94 |
>
> We acknowledge the limited baselines for object and style concepts and will include more in our final paper.
> # Clarifying Median Estimation Independence Across Timesteps
> We want to clarify that the conditional text embeddings at each timestep are **independent** of one another. For *each timestep*, we sample noise from Gaussian distribution and apply adaptive median smoothing to obtain the conditional text embedding. This smoothed text embedding is then input into the U-Net for the current denoising step. Although the computation of the smoothed text embedding occurs at each timestep, it can be performed in parallel, making it more efficient than computing the median U-Net noise prediction, as shown in Table 2 of our manuscript.
> # Theoretical Tightness under Large Perturbation
> The bound in Equation (2) remains theoretically meaningful even for large adversarial perturbation, as the bound explicitly depends on the interplay between perturbation norm upper bound $\rho$ and the **noise magnitude** $\sigma$ used in median smoothing.
>
> According to Lemma 3.1, the perturbed output $\mathcal{E}\_{0.5}(y+\delta)$ is bounded by $\underline{\mathcal{E}}\_{\underline{p}}(y)$ and $\overline{\mathcal{E}}_{\overline{p}}(y)$. Tighter bounds are achieved when $\underline{p}$ and $\overline{p}$ are closer to 0.5. The probabilities $\underline{p}:=\Phi\left(-\frac{\rho}{\sigma}\right)$ and $\overline{p}:=\Phi\left(\frac{\rho}{\sigma}\right)$ suggest that increasing $\sigma$ can tighten the bound when $\rho$ is large. Thus, **theoretically**, the bound remains effective even with large perturbations.
>
> In **practice**, while $\rho$ may be large, increasing $\sigma$ helps maintain adversarial robustness. However, excessive $\sigma$ may harm the model utility, so it requires careful tuning.
>
> # Reasons Behind Choosing ESD and UCE as Base Unlearned Models
> We selected ESD and UCE as the base unlearned models for the following reasons:
>
> - **Widely Recognized Baselines**: ESD and UCE are frequently used in the field of unlearning for diffusion models.
> - **Distinct Paradigms**: They represent two primary diffusion unlearning paradigms: ESD is associated with fine-tuning, while UCE pertains to model editing.
> - **Ensuring Fair Comparison**: The adversarial erasing-based baselines, RACE and RECE, are implemented using ESD and UCE, respectively. Therefore, selecting ESD and UCE as the base unlearned models ensures a fair comparison between our method and adversarial erasing-based approaches.
> # Discussion on AdvUnlearn
> We will cite and discuss AdvUnlearn [1] in our paper. The detailed discussion is provided in our response to Reviewer ZpJv.
>
> [1] Defensive Unlearning with Adversarial Training for Robust Concept Erasure in Diffusion Models, NeurIPS 2024.
> # Notation Section
> Thanks for your advice. We will include a notation section in our paper.

---

### Official Review · Reviewer_9A3r · 2025-03-18

**Overall Recommendation:** 3

**Summary:**

Even after unlearning, models are still vulnerable to adversarial inputs that can expose users to inappropriate contents. Existing adversarial defense methods still have difficulty with balancing the adversarial robustness and the generation quality. To address these issues, the paper proposes an inference-time defense strategy to defend against adversarial text prompts. To do so, the paper formulates the optimization for robustness as a robust regression problem, which is extended to a generalized median smoothing framework incorporating anisotropic noise. Finally, the paper proposes a token-wise Adaptive Median Smoothing strategy that applies noise of intensity that is dynamically adjusted according to the relevance of tokens to target concepts.

**Claims And Evidence:**

Claims are reasonable.

**Essential References Not Discussed:**

N/A

**Experimental Designs Or Analyses:**

Yes, the experimental designs and analyses seem sound and valid.

**Methods And Evaluation Criteria:**

Yes, I couldn't find any issue.

**Other Comments Or Suggestions:**

Typos/grammar erros
- line 229: The findings in Theorem 3.2 suggests -> The findings in Theorem 3.2 suggest
- line 241: we propose a Efficiency -> we propose an Efficiency
- run-on sentences in line 317-319

**Other Strengths And Weaknesses:**

Strengths:
- It’s a reasonable and nice perspective to view adversarial defense as robust regression problem
- This robust regression perspective allows the proposed method to be robust to adversarial attacks by employing a median smoothing framework.
- The paper does not stop at just applying a median smoothing framework, but finds the problem with applying it naively to concept erasure framework. Thus, the paper proposes a generalized median smoothing framework with anisotropic noise.


Weakness:
- If relevance score relies on token similarity, I'm guessing that it can be vulnerable to vague adversarial prompts (e.g., prompts that have no nudity-related words but lead to nudity generation). It would be great if the paper shows the analysis on this with various vague adversarial prompts .
- It seems that hyperparameters need to be finetuned for each baseline model. How robust is the proposed framework to the choice of hyperparameters? It would be great to show this analysis. The ablation study shows the results for only one baseline.
- To show how robust the proposed framework is to adversarial attacks, the paper could improve with the analysis that shows when the proposed framework fails and succeeds.
- The paper has a few typos/grammar errors. A thorough proof-reading is recommended.
- There is no detail on positive & negative prompts. Furthermore, which model has used positive & negative prompt list from ViSU? I think it's unfair to compare against models that did not have access to this list. The paper should provide an ablation study on the proposed method without prompt list from ViSU. Also, the paper should compare against existing methods that employ the list from ViSU. The paper should also use the prompt list to re-train methods that do not employ the prompt list and compare against them as well.

**Questions For Authors:**

Questions are written in the weakness section.

**Relation To Broader Scientific Literature:**

Formulating adversarial-robustness of concept erasure as robust regression problem and extending it to a generalized median smoothing framework brings non-trivial contribution and new insights to the field.

**Theoretical Claims:**

Yes, I couldn't find issues.

---

> ### Author Rebuttal · Authors · 2025-03-31
>
> # Analysis of Vague Adversarial Prompts
>
> Thanks for your insightful comment. Our method’s robustness against vague adversarial prompts is validated through evaluations on the **MMA attack** [1], which constructs adversarial prompts avoiding sensitive words while inducing unsafe generations. As shown in Table 1 of our manuscript, our method reduces the attack success rate (ASR) for MMA from $7.50\\%$ to $4.23\\%$ (ESD model) and from $39.30\\%$ to $17.53\\%$ (UCE model), demonstrating effectiveness against such vague adversarial prompts.
>
> [1] MMA-Diffusion: MultiModal Attack on Diffusion Models, CVPR 2024.
>
> # Ablation Study of Hyper-Parameters
>
> In response, we have conducted an ablation study on hyper-parameters using the other unlearned model, **UCE**. The updated results, provided [here](https://anonymous.4open.science/r/Re-ML/tb4.png), indicate that moderately increasing $\sigma_0$ and $k$ enhances the adversarial robustness of unlearned models without significantly compromising their utility.
>
> # Discussion on Failure Cases
>
> We have provided qualitative results [here](https://anonymous.4open.science/r/Re-ML/fig5.png). These results reveal that the framework occasionally fails on "hard cases" where adversarial prompts bypass default hyper-parameters ($\sigma_0$), allowing unsafe concept restoration (columns 2 & 5). While increasing $\sigma_0$ mitigates such cases, it risks degrading benign-generation quality. This highlights a trade-off between robustness and utility under fixed hyper-parameters. Future work will explore dynamic $\sigma_0$-adjustment based on input prompt to better balance these objectives.
>
> # Clarifying the Use of Prompt Pairs from ViSU
>
> Thank you for your valuable feedback. Here is our response:
>
> - **Details on Prompts**: As reported in [2], the authors fine-tuned a large language model (LLM), Llama 2, to generate unsafe sentences from safe ones collected from COCO captions. The training set contains 7,863 prompt pairs for the nudity concept.
> - **ViSU for Baseline Methods**: Most defense baselines in our manuscript do not require *prompt pairs*. Specifically, pre-inference defenses (RACE and RECE) *optimize* adversarial prompts and subsequently erase them. For training-free defenses, SLD uses *inappropriate keywords* to calculate unsafe guidance, while Blacklist uses *NSFW words* for filtering.
> - **Ablation Without ViSU**: To further demonstrate the effectiveness of our proposed method, we implemented "*nudity*" unlearning without using prompt pairs from ViSU. Specifically, we first used an uncensored LLM interface on Hugging Face to generate 100 nudity-related prompts (positive prompts) and then used another LLM interface (e.g., Kimi) to transform them into safe versions (i.e., negative prompts). We then used these 100 prompt pairs (denoted as **Self-Gen**) in our experiments, with results shown in the table below (a detailed table with attack success rates for each attack is provided [here](https://anonymous.4open.science/r/Re-ML/tb1.png)). Overall, our method remains effective even with just 100 collected prompt pairs. In the detailed table, we observe that adversarial robustness against vague adversarial prompts (i.e., MMA) degrades, suggesting that collecting more prompt pairs further enhances defense against vague adversarial prompts.
> | Unlearned Model | Defense Methods | ASR $\downarrow$ | FID $\downarrow$ | CLIP $\uparrow$ |
> | --------------- | --------------- | ---------------- | ---------------- | --------------- |
> | ESD             | w/o defense     | 25.42            | 7.161            | 30.18           |
> | ESD             | Ours (ViSU)     | 6.80             | 7.365            | 30.10           |
> | ESD             | Ours (Self-Gen) | 5.33             | 7.436            | 30.09           |
> | UCE             | w/o defense     | 40.34            | 4.379            | 31.02           |
> | UCE             | Ours (ViSU)  | 11.02            | 5.343            | 30.48           |
> | UCE             | Ours (Self-Gen) | 15.93            | 5.526            | 30.54           |
> - **Generalization to Other Concepts**: ViSU primarily contains unsafe prompts (e.g., nudity and violence) and is not suitable for other types of concepts (e.g., objects and styles). For these concept types, we first used an LLM interface to generate 100 positive prompts depicting the target concept, and then instructed the LLM to remove target concept-related elements to create negative prompts. The experimental results for these additional concept types are provided in the `Evaluation Across Object and Style Concepts` section in our response to Reviewer CcZX.
>
> [2] Safe-CLIP: Removing NSFW Concepts from Vision-and-Language Models, ECCV 2024.
>
> # Typos & Grammar Errors
>
> Thank you for your careful review. We have corrected the identified typos/grammar errors and will perform a thorough proofread of the entire manuscript to ensure clarity and correctness.

---

> > ### Comment · Reviewer_9A3r · 2025-04-04
> >
> > Thank you for the rebuttal, which has addressed most of my concerns, but not all of them.
> > Particularly, for the ablation on ViSU, I think baseline models, such as ESD, UCE, and Concept Ablation, can be trained with pairs of positive and negative prompts. Can't you take positive prompts as target concepts and negative prompts as anchor concepts? The proposed model's reliance on pairs of positive and negative prompts comes to me as weakness of the proposed framework. With the knowledge of the list of positive and negative prompts is available to adversaries, adversarial prompts could be easily generated to bypass the framework.
> > But, considering the balance between strengths and weakness of the proposed framework, I retain the score.

---

> > > ### Author Response · Authors · 2025-04-06
> > >
> > > We appreciate your concerns and address them as follows:
> > > # ESD and UCE with ViSU Prompt Pairs
> > >
> > > We have incorporated ViSU into the training processes of ESD and UCE, as detailed below:
> > >
> > > - **Implementation:** In the **official implementations** of ESD and UCE, **keywords** (e.g., "*nudity*") are typically used for target concepts, while null text ("") serves as anchor concepts. We acknowledge that positive prompts can be adapted as targets and negative prompts as anchors in ESD and UCE training. Specifically, for ESD, we randomly sample prompt pairs for fine-tuning in each iteration. For UCE, all prompt pairs are used for model parameter editing.
> > >
> > > - **Experimental Results**: We evaluated adversarial robustness specifically targeting the "*nudity*" concept through six attacks and calculated the average attack success rate (ASR). Model utility was assessed using FID and CLIP scores. The results are presented in the table below (a detailed table with attack success rates for each attack is provided [here](https://anonymous.4open.science/r/Re-ML/tb6.png)).
> > > | Unlearned Model | ViSU | Defense Methods | ASR $\downarrow$ | FID $\downarrow$ | CLIP $\uparrow$ |
> > > | --------------- | ---- | --------------- | ---------------- | ---------------- | --------------- |
> > > | ESD             | ❌    | w/o defense     | 25.42            | 7.161            | 30.18           |
> > > | ESD             | ✔️    | w/o defense     | 22.62            | 6.741            | 30.31           |
> > > | ESD             | ✔️    | Ours            | 6.80             | 7.365            | 30.10           |
> > > | UCE             | ❌    | w/o defense     | 40.34            | 4.379            | 31.02           |
> > > | UCE             | ✔️    | w/o defense     | 0.00             | 167.569          | 20.41           |
> > > | UCE             | ✔️    | Ours            | 11.02            | 5.343            | 30.48           |
> > >
> > > For **ESD**, incorporating ViSU prompt pairs does not negatively impact model utility, but the **adversarial robustness improvement is slight** (ASR: $25.42\\%\rightarrow22.62\\%$). Unlike ESD, which relies on fine-tuning, **UCE** is based on model editing. Incorporating a large number of prompt pairs can lead to excessive modifications to the model's parameters, which may **significantly harm its utility** (FID: $4.38\rightarrow167.57$). However, our method, which employs ViSU to compute relevance scores for adaptive median smoothing during inference, **effectively enhances adversarial robustness while preserving model utility**. As demonstrated in our previous response, even without ViSU, using our self-generated prompt pairs (**Self-Gen**) achieves comparable performance.
> > >
> > > # Prompt List for Deployment and Its Vulnerabilities under Attack
> > >
> > > Thanks for your insightful comment. Our response is as follows:
> > >
> > > Firstly, we want to clarify that prompt pairs are used to calculate the **concept vector** for the target concept, which is then utilized to compute the **relevance score**. This score determines the **noise intensity** for each token. Our previous ablation study of ViSU demonstrated that **Self-Gen** prompt pairs, generated using the LLM interface, are also effective.
> > >
> > > Although ViSU is publicly available and accessible to attackers, defenders can generate similar prompt pairs (like **Self-Gen**) using the LLM interface. The inherent **randomness** in the prompt generation process makes it difficult for attackers to replicate the exact prompt list used by defenders, thus making bypass attempts **more challenging**. Even if attackers were to obtain the exact list, designing an attack strategy to circumvent our framework would require careful planning, as, to our knowledge, existing attack methods are **not** equipped to handle such scenarios. Future efforts will focus on examining sophisticated attack techniques that might penetrate our framework, thus allowing us to bolster its robustness.
> > >
> > > We sincerely thank you again for your valuable feedback and suggestions, which have greatly assisted us in improving the quality of our paper.

---

### Decision · Program_Chairs · 2025-05-01

**Decision:**

Accept (poster)

**Comment:**

This paper presents a defense method against adversarial inputs for unlearned text-to-image diffusion models. The main idea is based on median smoothing, which adaptively adds noise to tokens based on the relevance score of the target concepts. The paper provides theoratical analysis from a regression perspective and the experiments that outperforms several earlier methods.

Reviewers generally agree that the paper is clear (FnEb, ZpJv), and that the idea is well-motivated (9A3r, FnEb) and novel for unlearning (9A3r, CzZX, ZpJv). However, they also pointed out some issues, such as the defense method only addressing specific cases like nudity and violence (CzZX), limited evaluation on unlearned T2I models (CzZX, FnEb), missing baseline (ZpJv), and unclear implementation details (9A3r, CzZX, FnEb).

After the rebuttal stage, most concerns have been dealt with by the additional results and clarifications. Although one reviewer did not respond after the rebuttal, potentially due to time conflict, most reviewers lean toward accepting the paper. The AC has read both the reviewers' comments and the authors' responses. AC feels the rebuttal has sufficiently addressed the concerns and recommends accepting the paper and also suggests that the authors include the extra discussion and results in the final version.